# Impact of the COVID-19 pandemic on breast cancer screening indicators in a Spanish population-based program: a cohort study

Guillermo Bosch[1,2], Margarita Posso[1,3]*, Javier Louro[1,3], Marta Roman[1,3], Miquel Porta[4,5,6], Xavier Castells[1,3,4], Francesc Macià[1,3]

[1]Department of Epidemiology and Evaluation,IMIM (Hospital del Mar Medical Research Institute), Barcelona, Spain; [2]Preventive Medicine and Public Health Training Unit PSMar-ASPB-UPF, Barcelona, Spain; [3]Research Network on Chronicity, Primary Care and Health Promotion (RICAPPS), Barcelona, Spain; [4]Universitat Autònoma de Barcelona, Barcelona, Spain; [5]Hospital del Mar Institute of Medical Research (IMIM PSMar), Barcelona, Spain; [6]Spanish Consortium for Research on Epidemiology and Public Health (CIBERESP), Madrid, Spain

## Abstract

**Background:** To assess the effect of the COVID-19 pandemic on performance indicators in the population-based breast cancer screening program of Parc de Salut Mar (PSMAR), Barcelona, Spain.

**Methods:** We conducted a before-and-after, study to evaluate participation, recall, false positives, the cancer detection rate, and cancer characteristics in our screening population from March 2020 to March 2021 compared with the four previous rounds (2012–2019). Using multilevel logistic regression models, we estimated the adjusted odds ratios (aORs) of each of the performance indicators for the COVID-19 period, controlling by type of screening (prevalent or incident), socioeconomic index, family history of breast cancer, and menopausal status. We analyzed 144,779 invitations from 47,571women.

**Results:** During the COVID-19 period, the odds of participation were lower in first-time invitees (aOR = 0.90 [95% CI = 0.84–0.96]) and in those who had previously participated regularly and irregularly (aOR = 0.63 [95% CI = 0.59–0.67] and aOR = 0.95 [95% CI = 0.86–1.05], respectively). Participation showed a modest increase in women not attending any of the previous rounds (aOR = 1.10 [95% CI = 1.01–1.20]). The recall rate decreased in both prevalent and incident screening (aOR = 0.74 [95% CI = 0.56–0.99] and aOR = 0.80 [95% CI = 0.68–0.95], respectively). False positives also decreased in both groups (prevalent aOR = 0.92 [95% CI = 0.66–1.28] and incident aOR = 0.72 [95% CI = 0.59–0.88]). No significant differences were observed in compliance with recall (OR = 1.26, 95% CI = 0.76–2.23), cancer detection rate (aOR = 0.91 [95% CI = 0.69–1.18]), or cancer stages.

**Conclusions:** The COVID-19 pandemic negatively affected screening attendance, especially in previous participants and newcomers. We found a reduction in recall and false positives and no marked differences in cancer detection, indicating the robustness of the program. There is a need for further evaluations of interval cancers and potential diagnostic delays.

**Funding:** This study has received funding by grants PI19/00007 and PI21/00058, funded by Instituto de Salud Carlos III (ISCIII) and cofunded by the European Union and Grant RD21/0016/0020 funded by Instituto de Salud Carlos III and by the European Union NextGenerationEU, Mecanismo para la Recuperación y la Resiliencia (MRR).

*For correspondence: mposso@parcdesalutmar.cat

Competing interest: The authors declare that no competing interests exist.

## Editor's evaluation

This paper will be of interest to public health specialists and cancer scientists working in cancer prevention. The work presents valuable data on how the COVID-19 pandemic has impacted breast cancer screening indicators compared with previous years. Overall, the results support the assertion that while many key indicators have not been substantially impacted, the screening participation rate declined compared to the pre-pandemic era.

## Introduction

In numerous health systems cancer screening programs were among the first activities interrupted by the COVID-19 pandemic after its irruption in early 2020. As reported in a survey by the International Cancer Screening Network, 97% of participating settings reported that COVID-19 had adversely impacted their screening programs, while 90% partially suspended their activity (*Puricelli Perin et al., 2021*; *World Health Organization, 2020*). Even in countries with notable success in containing the pandemic, like Taiwan, the population attending screening decreased during the first half of 2020 (*Peng et al., 2020*).

In Europe, breast cancer screening is mostly provided through organized programs offering routine mammography examination to women aged from 45–50 to 69–74 years. The programs follow the guidelines of the European Commission Initiative for Breast Cancer Screening and Diagnosis (*The European Commission's science and knowledge service, 2020*). These guidelines recommend an evidence-based set of performance indicators to evaluate the quality of the screening provision (*Muratov et al., 2020*). The suspension of these programs led to a reduction in cancer diagnoses. For instance, in the Netherlands and Austria, the number of breast cancer diagnoses decreased substantially and remained lower than expected until screening was rebooted (*Dinmohamed et al., 2020*; *Tsibulak et al., 2020*), while in Italy, between January and May 2020, 53% fewer screens were performed, with a median delay of 2.7 months for screening mammograms (*Mantellini et al., 2020*).

Currently, the evidence of the effect of the COVID-19 pandemic on breast cancer screening performance indicators has been mostly provided by simulation models and longitudinal studies are scarce. In Canada, Yong et al. used a mathematical model to estimate that a 3-month halt would have led to 664,000 fewer screening mammograms than expected, based on nationwide data from the previous year. It would also have decreased breast cancer diagnoses by 7% in 2020 and caused 110 excess deaths by 2029 (*Yong et al., 2021*). Similar models in Italy reported that 8125 breast cancer diagnoses were expected to be delayed due to a 3-month interruption of screening programs, representing 25% of the 32,500 yearly screening diagnoses nationwide (*Vanni et al., 2020*).

Spain was one of the first and most affected countries in Europe during the spring of 2020 (*Gallo et al., 2021*; *Karlinsky and Kobak, 2021*). On March 14, a general lockdown was enforced, and breast cancer screening was interrupted (*Alfonso Viguria and Casamitjana, 2021*). Restrictive measures were slowly withdrawn during the following 3 months until June 21, when the lockdown ended (*La Vanguardia, 2021*). To reintroduce the screening programs as soon as possible while continuing to control the risk of COVID-19 transmission, mammography centers established new safety guidelines (*Maio et al., 2021*; *Pediconi et al., 2020*).

Given the scarcity of longitudinal studies, we used a before-and-after design including data from a population-based program from 2012 to 2021. We aimed to assess the impact of the COVID-19 pandemic on the performance indicators of the program of Parc de Salut Mar (PSMAR) of Barcelona, Catalonia, Spain.

## Materials and methods

### Study design

In this before-and-after study, we compared the population-based breast cancer screening indicators obtained in a single population before and after the COVID-19 pandemic.

## Study population

In Spain, publicly funded mammographic screening for breast cancer is offered every 2 years to women aged 50–69 years (*Castells et al., 2008*). The screening examination at PSMAR consists of both a mediolateral oblique and a craniocaudal digital (two-dimensional) mammographic view of each breast. Two independent radiologists with extensive experience perform blinded double reading of mammograms. Disagreements are resolved by a third senior radiologist (*Posso et al., 2022*). The program covers the population of four districts of the city of Barcelona, with around 620,000 inhabitants, and approximately 75,000 eligible women. Until the pandemic, screening invitations were sent by postal mail with a prescheduled mammogram appointment to all invited women. Since 2020, previous participants and first-time invitees have been invited to participate by telephone, in addition to a previously sent letter informing them of the upcoming call. Previous nonparticipants still receive an invitation via postal mail, without a preset date, inviting them to call a specific telephone number to schedule the mammogram at a convenient time.

Invitations are issued during the 2-year duration of each screening round according to the geographical criteria set by Basic Health Areas (BHA), which are the basic territorial healthcare units of the city. In this analysis, we used data from 10 out of the 25 BHA covered by the PSMAR breast cancer screening program. The 10 BHA selected were those affected by the interruption and delay of the screening program during the first year of the pandemic, invitations for women from such BHA should have been sent out between March 2020 and March 2021.

For this study, the pre-COVID-19 period started in March 2012, ended in March 2019, and was divided into four pre-COVID-19 rounds of 2 years each. The post-COVID-19 period, therefore, went from March 2020 to March 2021, and included one screening round. We extended the follow-up until September 2021 to include the process of cancer diagnosis for women attending screening in the post-COVID round.

We obtained 144,779 observations, which are screening invitations linked to the following actions that may follow them (participation, recall, cancer detection), from 47,571 eligible women throughout the 10 years of the study. Each of these observations represented an invitation to the screening program. In our study population, age group, socioeconomic status, and type of screening round were statistically different in the post- and pre-COVID-19 periods (*Table 1*). The percentage of invited women living in high-income areas decreased slightly (−1.03%) as did that of women younger than 55 years (−1.83%). The distribution of the type of screening of invited women also changed, with a higher percentage of invitations for prevalent screening (+1.69%), especially first-time invitees (+2.90%).

## Data sources

All data from women eligible for screening were obtained from the management database of the program, which is updated yearly with the screening status and baseline characteristics of both participants and nonparticipants over the years. We completed the information on cancer histology and tumor stages with data the clinical and pathological records.

Each screening invitation was considered an independent measurement. All measurements were pseudoanonymized by using an individual ID for each woman while removing all personal data. Therefore, although multiple measurements per woman could be obtained from invitations in different rounds, all of them shared the same pseudoanonymized ID.

## Outcomes and variables of interest

From the screening database of the program, we obtained the indicators on the selected BHA for the 2020–2021 screening round, which, as mentioned, was categorized as the post-COVID-19 period. We compared the post-COVID-19 indicators with those from the four previous screening rounds of the same BHA, categorized as the pre-COVID-19 period. We used five main indicators of the program: participation, recall, false positives, compliance with recall, and detection rate. In addition, we also compared the following characteristics of the detected tumors, histology (invasive vs. in situ), tumor size, lymphatic invasion, the presence of metastases, and stage at diagnosis.

Participation was measured as the percentage of women invited for screening who underwent mammography in the corresponding round. Invited women were those fulfilling the selection criteria (age 50–69 years, residence in the selected BHA) and who did not meet of the exclusion criteria of the program. The main exclusion criteria were a change of address outside the geographic area of

**Table 1.** Baseline characteristics of women invited to the PSMAR breast cancer screening program in the BHA (Basic Health Areas) affected by the COVID-19 pandemic in the area of reference of Parc de Salut Mar (PSMAR), in Barcelona, Spain in the period 2012–2021 by screening round.

| | | Pre-COVID 4 round (2012–2013) | | Pre-COVID 3 round (2014–2015) | | Pre-COVID 2 round (2016–2017) | | Pre-COVID 1 round (2018–2019) | | Pre-COVID total (2012–2019) | | Post-COVID +1 round (2020–2021) | | Difference post-pre | Dif |
|---|---|---|---|---|---|---|---|---|---|---|---|---|---|---|---|
| | | n | % | n | % | n | % | n | % | n | % | n | % | | |
| **Age group** | 50–54 | 9103 | 31.79 | 9495 | 32.55 | 9451 | 32.25 | 9013 | 31.41 | 37,062 | 32.00 | 8743 | 30.17 | -1.83 | < |
| | 55–59 | 7153 | 24.98 | 7345 | 25.18 | 7569 | 25.83 | 7631 | 26.59 | 29,698 | 25.65 | 7899 | 27.26 | 1.61 | < |
| | 60–64 | 6687 | 23.36 | 6573 | 22.53 | 6692 | 22.84 | 6666 | 23.23 | 26,618 | 22.99 | 6831 | 23.57 | 0.59 | < |
| | 65–70 | 5688 | 19.87 | 5756 | 19.73 | 5593 | 19.09 | 5387 | 18.77 | 22,424 | 19.36 | 5504 | 18.99 | -0.37 | |
| **Socioeconomic index level** | Low | 4458 | 15.57 | 4472 | 15.33 | 4371 | 14.92 | 4371 | 15.23 | 17,672 | 15.26 | 4536 | 15.65 | 0.39 | |
| | | 2303 | 8.04 | 2356 | 8.08 | 2329 | 7.95 | 2338 | 8.15 | 9326 | 8.05 | 2506 | 8.65 | 0.59 | < |
| | Middle-high | 15,209 | 53.12 | 15,668 | 53.71 | 15,864 | 54.13 | 15,557 | 54.21 | 62,298 | 53.80 | 15,600 | 53.84 | 0.04 | |
| | High | 6661 | 23.26 | 6673 | 22.88 | 6741 | 23.00 | 6431 | 22.41 | 26,506 | 22.89 | 6335 | 21.86 | -1.03 | < |
| **Type of screening** | | | | | | | | | | | | | | | |
| Prevalent | First invitation | 4409 | 15.40 | 4571 | 15.67 | 4271 | 14.57 | 3831 | 13.35 | 17,082 | 14.75 | 5114 | 17.65 | 2.90 | < |
| | Previous nonparticipant | 8497 | 29.68 | 8639 | 29.62 | 9049 | 30.88 | 9012 | 31.40 | 35,197 | 30.39 | 8456 | 29.18 | -1.21 | < |
| | Total | 12,906 | 45.08 | 13,210 | 45.29 | 13,320 | 45.45 | 12,843 | 44.75 | 52,279 | 45.15 | 13,570 | 46.83 | 1.69 | |
| Incident | Regular participant | 13,410 | 46.84 | 13,478 | 46.21 | 13,495 | 46.05 | 13,390 | 46.66 | 53,773 | 46.44 | 13,070 | 45.10 | -1.33 | < |
| | Irregular participant | 2315 | 8.09 | 2481 | 8.51 | 2490 | 8.50 | 2464 | 8.59 | 9750 | 8.42 | 2337 | 8.07 | -0.35 | |
| | Total | 15,725 | 54.92 | 15,959 | 54.71 | 15,985 | 54.55 | 15,854 | 55.25 | 63,523 | 54.85 | 15,407 | 53.17 | -1.69 | |
| **Total** | | 28,631 | 100.00 | 29,169 | 100.00 | 29,305 | 100.00 | 28,697 | 100.00 | 115,802 | 100.00 | 28,977 | 100.00 | | |

Pre-COVID totals calculated as the sum of the four pre-COVID screening rounds. Percentages show the distribution for the columns of each variable. p values obtained with chi-square test comparing the total pre- and post-COVID proportions <0.001 for all variables. Dif = '^' indicates a statistically significant difference of $p < 0.05$ between columns of the respective category.

the program, previous breast cancer, high hereditary risk of breast cancer and errors in identification or personal data.

Three other outcomes were analyzed using only the screening participants. The recall rate was estimated as the percentage of participants who were advised to undergo further assessment to rule out malignancy, whether noninvasive or invasive (ultrasound, tomosynthesis, contrast-enhanced mammography, biopsy, and/or others). False positives were estimated based on the percentage of women who underwent additional noninvasive or invasive assessments but who did not have a diagnosis of cancer after completion of additional examinations. The detection rate was the number of breast cancers detected at screening per 1000 participants. We calculated this rate, stratifying by type of breast cancer histology (i.e., the invasive or in situ cancer detection rate). Finally, compliance with recall was analyzed only among patients advised to undergo further assessment, the percentage of these patients who agreed to take additional tests in our facilities.

We stratified all the invitations by type of screening between prevalent or incident screening. Prevalent screening refers to the process of inviting women who have never participated in screening, while incident screening refers to inviting previous participants. In terms of prevalent screening, we differentiated between first-time invitees, and nonparticipants, referring to previously invited women who had never participated. For incident screening, we differentiated between previous participants who had participated in the previous round (regular participants) and those not participating in the last round (irregular participants).

We categorized women according to their age at the time of invitation in four groups: 50–54, 55–59, 60–64, and 65–70 years old. Socioeconomic status was estimated with a compound socioeconomic index, created by the Government of Catalonia to assign resources to primary healthcare, based on the index of each BHA (*Agència de Qualitat i Avaluació Sanitàries de Catalunya, 2017*). Each woman was assigned the socioeconomic index of the BHA where she was living. Higher values denote a lower socioeconomic level.

We evaluated clinical variables such as menopausal status and family history of breast cancer. Breast cancer histology differentiates between in situ and invasive tumors. According to the TNM Breast Cancer 8th Edition classification (*Giuliano et al., 2017*), tumor size was measured in millimeters, lymphatic invasion as the extension of malignant cells, metastasis as its presence or absence, and stage at diagnosis as I, II, III, and IV TNM categories. We used the pathology (p)TNM preferably, and only used the clinical (c)TNM for women with neoadjuvant treatment (*Román et al., 2017*). Other epidemiological and clinical variables such as educational level or history of hormone replacement therapy were not included in the analyses due to a high percentage of missing values (>10%).

## Statistical analysis

We first compared the characteristics of the invited population among the different screening rounds to describe variations in their distribution. We evaluated differences in the categories using the chi-square test or the exact Fisher's test when appropriate.

Then, we created multilevel logistic regression models to estimate adjusted odds ratios (aORs) of each of the performance indicators and their corresponding 95% confidence intervals (95% CI) for the COVID-19 period, adjusting by the clinically relevant variables.

For participation, we included the following variables in the model: type of screening round (prevalent vs. incident), age group, and socioeconomic index. We found a strong interaction between COVID-19 and the type of screening round. Therefore, we created a new variable, which represented this interaction. Hence, the final models for participation differentiated four screening groups (prevalent-first-time invitee, prevalent-previous nonparticipant, incident-regular participant, and incident-irregular participant). We obtained crude results and adjusted by age and socioeconomic index. For compliance with recall, we used a logistic regression model to obtain crude odds ratios since we did not adjust for any variables due to the reduced sample size.

We created three additional models, including only participants, to assess the impact of COVID-19 on the other main indicators of the screening program: recall and false positives. Finally, we used independent logistic regression models for the screen-detected cancer rate (invasive or in situ). These models were adjusted for age group, menopausal status, and breast cancer family history.

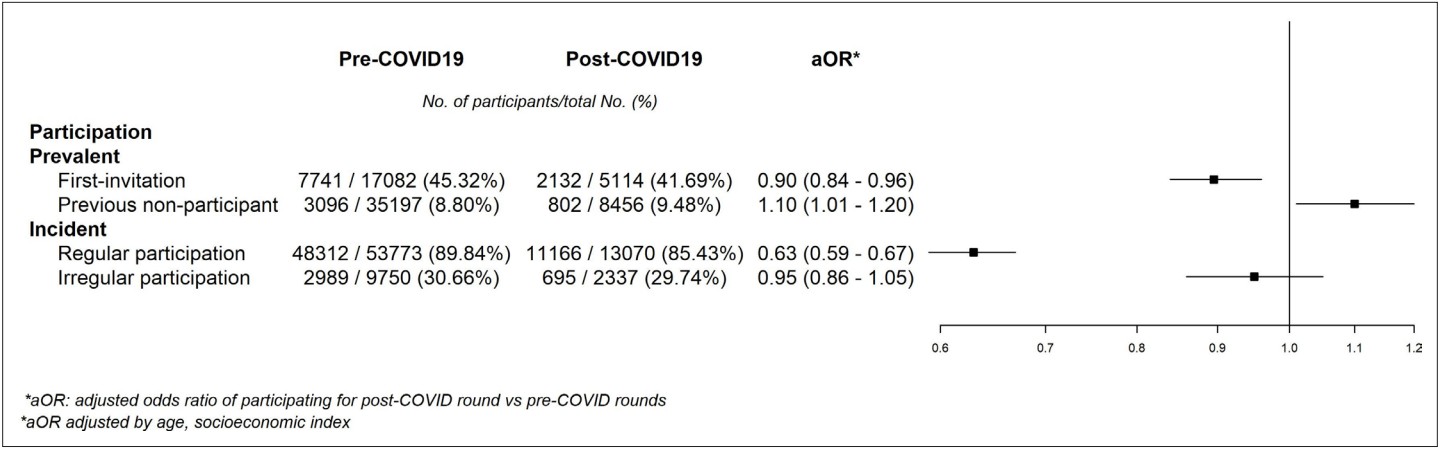

**Figure 1.** Adjusted odds ratios of pre–post COVID-19 models for participation.

Finally, we compared the stage at diagnosis and the remaining cancer characteristics (size, lymph node invasion, and metastasis invasion) of cases detected in the screening program in the pre- and post-COVID-19 periods.

Statistical tests were two sided and all p values lower than 0.05 were considered statistically significant.

SPSS version 25 software was used for the creation and validation of the database and recodification of variables, while statistical software R version 3.5.0 (Development Core Team, 2014) was used for the logistic regression models.

## Ethical aspects

The study guaranteed Spain's legal regulations on data confidentiality (law 15/99 of December 13 on the protection of personal data). Due to the retrospective nature of the study and the absence of direct contact with women, which did not affect their relationship with the program, informed consent was waived by the Ethics Committee of PSMAR, which approved the study (reg. 2021/9866).

## Results

Participation in the program was affected differently depending on the type of screening. The aOR of participation between the post- and pre-COVID-19 periods was 0.90 (95% CI = 0.84–0.96) for the group of first-time invitees. The aOR was 1.10 (95% CI = 1.01–1.20) for the previous nonparticipant group between the post- and pre-COVID-19 periods. For the group of women who had participated in the previous round (regular participants), the aOR of participation was 0.63 (95% CI = 0.59–0.67), and for those not participating in the last round (irregular participants), the aOR was 0.95 (95% CI = 0.86–1.05) (*Figure 1*).

We also found statistically significant differences in the distribution of baseline characteristics of participants during the post-COVID-19 round compared with the mean distributions of the pre-COVID-19 period (*Table 2*). The percentage of participants younger than 55 years decreased (−1.64%) but the percentage aged between 55 and 59 years increased (+1.73%). The percentage of participants from high socioeconomic level areas slightly decreased (−0.62%). The biggest changes in the distribution of participants were seen between types of screening, with a substantial decrease among participants in the incident screening group (−2.39%) and an increase in the percentage of prevalent screening, especially first-time invitees (+1.96%). The percentage of participants with a family history of breast cancer increased by 2.53%.

Analysis of participation proportions in different groups according to their characteristics revealed that participation decreased with age, with the largest reduction in participation occurring in women older than 65 years (−3.65%). Although participation decreased among all socioeconomic levels, the decrease was greater in those with middle-low (−6.03%) and low (−5.61%) status. Participation was greatly reduced among regular participants, who had participated in the previous round, with a 4.41%

**Table 2.** Baseline characteristics of participants of the PSMAR breast cancer screening program in the BHA affected by the COVID-19 pandemic in the area of reference of Parc de Salut Mar (PSMAR), in Barcelona, Spain in the period 2012–2021 by screening round.

| Characteristic | Category | Pre-COVID 4 round (2012–2013) n | % | Pre-COVID 3 round (2014–2015) n | % | Pre-COVID 2 round (2016–2017) n | % | Pre-COVID 1 round (2018–2019) n | % | Pre-COVID total (2012–2019) n | % | Post-COVID +1 round (2020–2021) n | % | Difference post–pre | Dif |
|---|---|---|---|---|---|---|---|---|---|---|---|---|---|---|---|
| Age group | 50–54 | 4550 | 29.22 | 4596 | 29.50 | 4546 | 29.39 | 4434 | 28.56 | 18,126 | 29.17 | 4073 | 27.53 | −1.64% | < |
| | 55–59 | 3912 | 25.12 | 3901 | 25.04 | 3985 | 25.77 | 4089 | 26.34 | 15,887 | 25.57 | 4038 | 27.29 | 1.73% | < |
| | 60–64 | 3784 | 24.30 | 3729 | 23.94 | 3710 | 23.99 | 3780 | 24.35 | 15,003 | 24.14 | 3664 | 24.77 | 0.62% | |
| | 65–70 | 3325 | 21.35 | 3352 | 21.52 | 3225 | 20.85 | 3220 | 20.74 | 13,122 | 21.12 | 3020 | 20.41 | −0.71% | |
| Socioeconomic index level | Low | 3282 | 21.08 | 3223 | 20.69 | 3201 | 20.70 | 3226 | 20.78 | 12,932 | 20.81 | 3065 | 20.72 | −0.10% | |
| | Middle-low | 1603 | 10.29 | 1612 | 10.35 | 1620 | 10.47 | 1630 | 10.50 | 6465 | 10.40 | 1586 | 10.72 | 0.31% | |
| | Middle-high | 8363 | 53.71 | 8537 | 54.80 | 8508 | 55.01 | 8626 | 55.57 | 34,034 | 54.77 | 8164 | 55.18 | 0.41% | |
| | High | 2323 | 14.92 | 2206 | 14.16 | 2137 | 13.82 | 2041 | 13.15 | 8707 | 14.01 | 1981 | 13.39 | −0.62% | < |
| Type of screening | First invitation | 2058 | 13.22 | 1957 | 12.56 | 1876 | 12.13 | 1850 | 11.92 | 7741 | 12.46 | 2133 | 14.42 | 1.96% | < |
| | Previous nonparticipant | 812 | 5.21 | 771 | 4.95 | 750 | 4.85 | 763 | 4.92 | 3096 | 4.98 | 802 | 5.42 | 0.44% | < |
| Prevalent | | 2870 | 18.43 | 2728 | 17.51 | 2626 | 16.98 | 2613 | 16.83 | 10,837 | 17.44 | 2935 | 19.84 | 2.40% | |
| | Regular participant | 12,003 | 77.09 | 12,116 | 77.78 | 12,095 | 78.20 | 12,098 | 77.94 | 48,312 | 77.75 | 11,166 | 75.47 | −2.28% | < |
| | Irregular participant | 698 | 4.48 | 734 | 4.71 | 745 | 4.82 | 812 | 5.23 | 2989 | 4.81 | 695 | 4.70 | −0.11% | |
| Incident | Total | 12,701 | 81.57 | 12,850 | 82.49 | 12,840 | 83.02 | 12,910 | 83.17 | 51,301 | 82.56 | 11,861 | 80.17 | −2.39% | < |
| Menopause | Yes | 12,069 | 77.52 | 12,358 | 79.34 | 12,212 | 78.98 | 12,267 | 79.03 | 48,906 | 78.71 | 11,683 | 78.97 | −0.18% | < |
| | No | 1986 | 12.76 | 1973 | 12.67 | 2095 | 13.54 | 2106 | 13.56 | 8160 | 13.13 | 1978 | 13.37 | 0.18% | |
| | Unknown | 1516 | 9.74 | 1247 | 8.00 | 1159 | 7.49 | 1150 | 7.41 | 5072 | 8.16 | 1134 | 7.66 | −0.11% | |
| Breast cancer family history | Yes | 2755 | 17.69 | 2915 | 18.71 | 3039 | 19.65 | 3213 | 20.70 | 11,922 | 19.19 | 3207 | 21.67 | 2.53% | < |
| | No | 12,792 | 82.15 | 12,633 | 81.10 | 12,390 | 80.11 | 12,265 | 79.01 | 50,080 | 80.59 | 11,532 | 77.94 | −2.53% | < |
| | Unknown | 24 | 0.15 | 30 | 0.19 | 37 | 0.24 | 45 | 0.29 | 136 | 0.22 | 56 | 0.38 | | |
| Education level | Primary or lower | 4641 | 29.81 | 4024 | 25.83 | 3370 | 21.79 | 2776 | 17.88 | 14,811 | 23.84 | 2032 | 13.73 | −11.78% | < |
| | Middle level | 6156 | 39.54 | 6367 | 40.87 | 6575 | 42.51 | 7371 | 47.48 | 26,469 | 42.60 | 7941 | 53.67 | 14.31% | < |
| | University | 2405 | 15.45 | 2658 | 17.06 | 2930 | 18.94 | 2777 | 17.89 | 10,770 | 17.33 | 2213 | 14.96 | −2.53% | < |
| | Unknown | 2369 | 15.21 | 2529 | 16.23 | 2591 | 16.75 | 2599 | 16.74 | 10,088 | 16.23 | 2609 | 17.63 | | |
| Total | | 15,571 | 100.00 | 15,578 | 100.00 | 15,466 | 100.00 | 15,523 | 100.00 | 62,138 | 100.00 | 14,795 | 100.00 | | |

Dif = '^' indicates a statistically significant difference of p<0.05 between columns of the respective category.

reduction, and slightly increased among previous nonparticipants, who participated for the first time despite having been previously invited (+0.69%) (*Table 3*).

Analysis of recall revealed modest decreases in the odds of being advised to undergo additional testing during the post-COVID-19 period in both the prevalent and the incident screening groups (aOR = 0.74 [95% CI = 0.56–0.99] and aOR = 0.80 [95% CI = 0.68–0.95]). The aOR of a false positive result for prevalent and incident screening was 0.92 (95% CI = 0.66–1.28) and 0.72 (95% CI 0.59–0.88), respectively. The aOR of cancer detection in the post-COVID vs. the pre-COVID-19 period was 1.01 (95% CI = 0.56–1.71) and 0.87 (95% CI = 0.63–1.17) in the prevalent and incident screening groups, respectively (*Figure 2*).

Compliance with recall did not significantly change in the post-COVID-19 round (OR = 1.26, 95% CI = 0.76–2.23), remaining stable with more than 97% of the women advised to undergo further assessment by taking additional diagnostic tests in our facilities (*Table 4*).

When differentiating between cancer histology, we found no statistically significant reductions in the odds of being diagnosed with a carcinoma in situ (aOR 0.74 [95% CI = 0.32–1.47]) or an invasive tumor (aOR 0.95 [95% CI = 0.70–1.26]), whereas the aOR for all tumors was 0.91 [95% CI = 0.69–1.18] (*Figure 3*). Finally, we observed no statistically significant differences in the distribution of the tumor size, lymphatic invasion, the presence of metastasis or stage at diagnosis between the pre- and post-COVID-19 periods. A statistically nonsignificant decrease of 4.47% in in situ tumors, and a nonsignificant increase of 4.95% in stages I were noted (*Table 5*).

None of the crude results significantly differed from the adjusted results (*Supplementary file 1*).

## Discussion

In this before-and-after study, we found that the pandemic reduced participation, but also that this impact differed according to each woman's history of participation. The frequency of recall for additional tests after mammography and the percentage of false-positive results were also significantly lower for incident screening, while for prevalent screening the reduction was only statistically significant for recall. Nevertheless, we found no significant differences in compliance with recall or cancer detection rate.

Women who became eligible for invitation to our population-based screening program for the first time in the post-COVID-19 period were significantly less likely to participate during the pandemic. This effect was also noted, and in a much higher degree, in women who had participated in the previous round. A reduction in participation, although nonstatistically significant, was also seen in women with previous irregular participation. Even though we could not identify the exact reasons behind the lower participation, we hypothesize that possible factors could be general insecurity related to attending hospitals, governmental restrictions of movement, fear of COVID-19 infection, and other uncertainties about the safety of participating in the screening process. This hypothesis is supported by data from a Danish study reporting that two of the reasons for postponing or canceling mammography appointments during the first year of the pandemic were fear and lack of clear guidance on the safety of screening (*Kirkegaard et al., 2021*).

Women who had been previously invited but had never attended our screening invitation seemed to participate slightly more during the pandemic period. The increase in participation was not expected since this group of women is that with the lowest participation in our setting (*Rodriguez et al., 1995*). However, this change could be explained by a plausible modification in attitudes to screening with a possible increase in health consciousness promoted by the pandemic, prompting women who had never been interested in screening to participate for the first time. Women who have never previously participated due to private screening may also have switched to the population-based program due to the effect of the pandemic on private clinics, which also had to stop their preventive care programs during lockdown.

Since the invitation process was adapted to the pandemic, the changes in participation could be related to the different strategies used to invite previous participants and nonparticipants. Nevertheless, previous research in our program showed that participation increased with invitation through direct contact with women (*Segura et al., 2001*), which could be comparable to telephone calls. This effect seemed to be especially relevant in low socioeconomic status areas, where there are more regular participants. Telephone reminders have also been proved to increase participation in different settings, although they usually follow a preset invitation date by postal mail (*Duffy et al., 2017*).

**Table 3.** Participation according to Baseline characteristics of participants of the PSMAR breast cancer screening program in the BHA affected by the COVID-19 pandemic in the area of reference of Parc de Salut Mar (PSMAR), in Barcelona, Spain in the period 2012–2021 by screening round.

| | | Pre-COVID 4 round (2012–2013) | | Pre-COVID 3 round (2014–2015) | | Pre-COVID 2 round (2016–2017) | | Pre-COVID 1 round (2018–2019) | | Pre-COVID total (2012–2019) | | Post-COVID +1 round (2020–2021) | | Difference post–pre | p valor |
|---|---|---|---|---|---|---|---|---|---|---|---|---|---|---|---|
| | | n | % | n | % | n | % | n | % | n | % | n | % | | |
| Age group | 50–54 | 4550 | 49.98 | 4596 | 48.40 | 4546 | 48.10 | 4434 | 49.20 | 18,126 | 48.91 | 4073 | 46.59 | -2.32% | <0.001* |
| | 55–59 | 3912 | 54.69 | 3901 | 53.11 | 3985 | 52.65 | 4089 | 53.58 | 15,887 | 53.50 | 4038 | 51.12 | -2.37% | <0.001* |
| | 60–64 | 3784 | 56.59 | 3729 | 56.73 | 3710 | 55.44 | 3780 | 56.71 | 15,003 | 56.36 | 3664 | 53.64 | -2.73% | <0.001* |
| | 65–70 | 3325 | 58.46 | 3352 | 58.23 | 3225 | 57.66 | 3220 | 59.77 | 13,122 | 58.52 | 3020 | 54.87 | -3.65% | <0.001* |
| | Low | 3282 | 73.62 | 3223 | 72.07 | 3201 | 73.23 | 3226 | 73.80 | 12,932 | 73.18 | 3065 | 67.57 | -5.61% | <0.001* |
| | Middle-low | 1603 | 69.60 | 1612 | 68.42 | 1620 | 69.56 | 1630 | 69.72 | 6465 | 69.32 | 1586 | 63.29 | -6.03% | <0.001* |
| | Middle-high | 8363 | 54.99 | 8537 | 54.49 | 8508 | 53.63 | 8626 | 55.45 | 34,034 | 54.63 | 8164 | 52.33 | -2.30% | <0.001* |
| Socioeconomic index level | High | 2323 | 34.87 | 2206 | 33.06 | 2137 | 31.70 | 2041 | 31.74 | 8707 | 32.85 | 1981 | 31.27 | -1.58% | 0.016* |
| Type of screening | First invitation | 2058 | 46.68 | 1957 | 42.81 | 1876 | 43.92 | 1850 | 48.29 | 7741 | 45.32 | 2133 | 41.71 | -3.61% | <0.001* |
| | Previous nonparticipant | 812 | 9.56 | 771 | 8.92 | 750 | 8.29 | 763 | 8.47 | 3096 | 8.80 | 802 | 9.48 | 0.69% | 0.05* |
| Prevalent | Total | 2870 | 22.24 | 2728 | 20.65 | 2626 | 19.71 | 2613 | 20.35 | 10,837 | 20.73 | 2935 | 21.63 | 0.90% | 0.024* |
| | Regular participant | 12,003 | 89.51 | 12,116 | 89.89 | 12,095 | 89.63 | 12,098 | 90.35 | 48,312 | 89.84 | 11,166 | 85.43 | -4.41% | <0.001* |
| | Irregular participant | 698 | 30.15 | 734 | 29.58 | 745 | 29.92 | 812 | 32.95 | 2989 | 30.66 | 695 | 29.74 | -0.92% | 0.38 |
| Incident | Total | 12,701 | 80.77 | 12,850 | 80.52 | 12,840 | 80.33 | 12,910 | 81.43 | 51,301 | 80.76 | 11,861 | 76.98 | -3.78% | <0.001* |
| Total | | 15,571 | 54.39 | 15,578 | 53.41 | 15,466 | 52.78 | 15,523 | 54.09 | 62,138 | 53.66 | 14,795 | 51.06 | -2.60% | <0.001* |

Pre-COVID totals calculated as the sum of the four pre-COVID screening rounds. Percentages show the proportion of participants among women invited for each category. p values obtained with chi-square test comparing the total pre- and post-COVID proportions. * indicates statistical significance, with a p-value < of equal to 0.05.

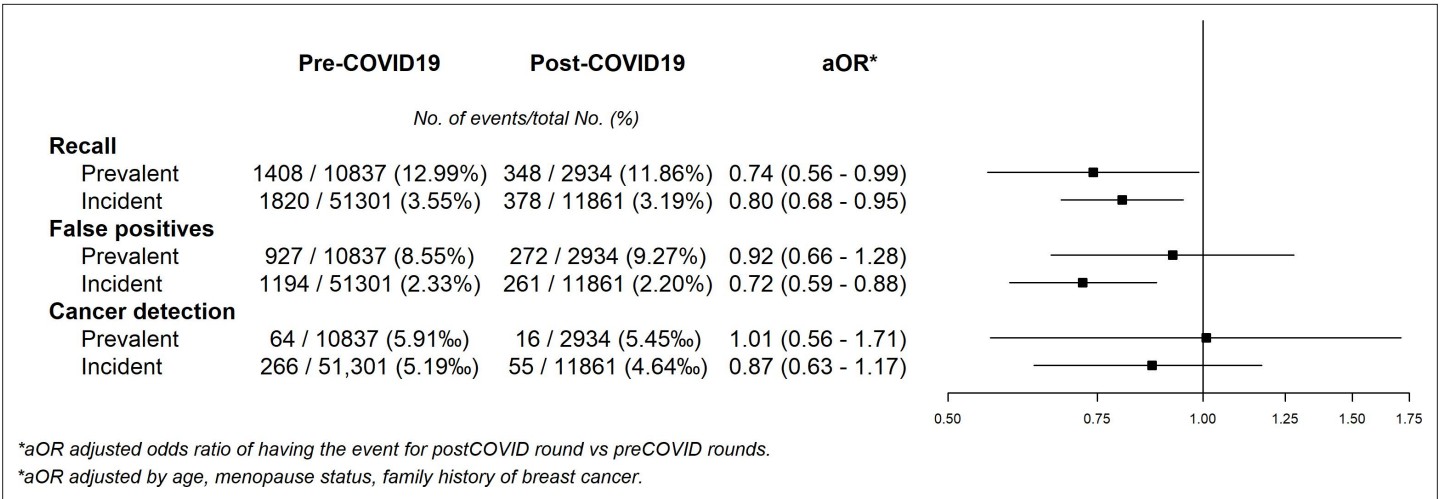

Figure 2. Adjusted odds ratios of pre–post COVID-19 models for recall, false positives, and cancer detection in participants.

Considering this evidence, we would assume that, without the telephone call, the decrease in participation could have been greater among regular participants. For previous nonparticipants, it is unlikely that an open invitation letter instead of a preset appointment could increase participation. Research in cervical cancer screening has proved that preset appointments increase participation in comparison to open invitations (*Lönnberg et al., 2016*), and therefore we believe that the increase in participation can be more feasibly explained by the previously discussed changes in private care and health consciousness.

Overall, our findings on participation adjusted by age and socioeconomic status showed that the effect of the pandemic on screening attendance depended on each woman's previous participation status. Although the aim of our study was not to evaluate the factors associated with participation, we found a lower representation of high-income women in the post-COVID-19 period, but this could probably be explained by the demographic changes in the invited population. Indeed, although participation decreased among all socioeconomic groups, this decline was greater in low-income areas. A systematic review of studies conducted before the pandemic reported lower participation in low-income groups, immigrants, nonhomeowners, and women with a previous false-positive result (*Mottram et al., 2021*). Furthermore, studies recently published in the United States have reported a decrease in participation, especially in underserved ethnic groups, with lower socioeconomic status, lack of insurance and longer travel time (*Amram et al., 2021*; *Miller et al., 2021*). Monitoring this information would allow programs to make efforts to promote participation among women at higher risk of not participating, especially under disruptive situations.

Despite the lower participation, the remaining performance indicators in our program did not seem to be negatively affected by the pandemic. Our results showed a statistically significant reduction in the recall rates of both prevalent and incident screening. These findings could be due to the increased workload caused by COVID-19 patients at our and many other hospitals, which strongly affected the radiology department in 2020 (*Posso et al., 2020*). We feared that repeat visits to the hospital might be perceived as increasing the risk of COVID-19 exposure, dissuading some women from undergoing

Table 4. Compliance with further assessment among patients assessed for recall in the BHAs affected by the COVID-19 pandemic in the area of reference of Parc de Salut Mar (PSMAR), in Barcelona, Spain from 2012 to 2021 by screening round.

| | | Pre-COVID −4 (2012–2013) | | Pre-COVID −3 (2014–2015) | | Pre-COVID −2 (2016–2017) | | Pre-COVID −1 (2018–2019) | | Pre-COVID total (2012–2019) | | Post-COVID (2020–2021) | | |
|---|---|---|---|---|---|---|---|---|---|---|---|---|---|---|
| | | n | % | n | % | n | % | n | % | n | % | n | % | OR (95% CI) |
| Compliance with recall | No | 36 | 4.57 | 32 | 3.08 | 11 | 1.42 | 10 | 1.61 | 89 | 2.76 | 16 | 2.20 | 1.26 (0.76–2.23) |
| | Yes | 752 | 95.43 | 1008 | 96.92 | 766 | 98.58 | 613 | 98.39 | 3139 | 97.24 | 710 | 97.80 | |

Pre-COVID totals were calculated as the sum of the four pre-COVID screening rounds. OR represent the crude odds ratios for compliance with further assessment for the post- vs. total pre-COVID-19 rounds.

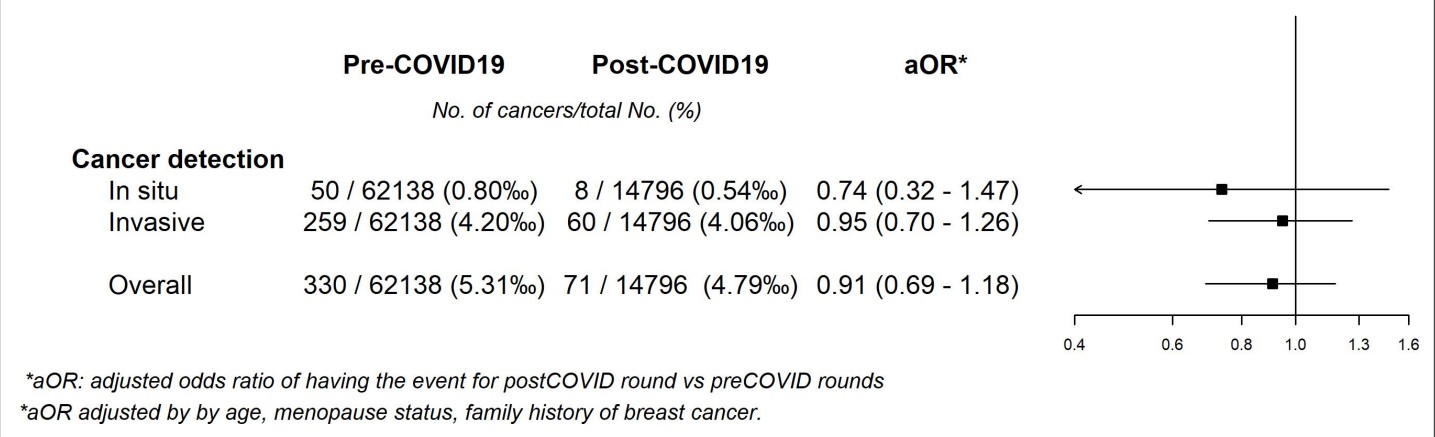

**Figure 3.** Adjusted odds ratios of pre–post COVID-19 models for cancer detection according to histology.

**Table 5.** Staging of cancers detected in each screening round in the BHA affected by the COVID-19 pandemic in the area of reference of Parc de Salut Mar (PSMAR), in Barcelona, Spain in the period 2012–2021 by screening round.

|  |  | Pre-COVID total (2012–2019) | | Post-COVID +1 round (2020–2021) | | Difference post–pre |
|---|---|---|---|---|---|---|
|  |  | n | % | n | % |  |
|  | In situ | 51 | 15.45 | 8 | 11.27 | −4.76% |
|  | 1 | 210 | 63.64 | 47 | 66.20 | 0.48% |
|  | 2 | 43 | 13.03 | 13 | 18.31 | 4.92% |
|  | 3 | 8 | 2.42 | 2 | 2.82 | 0.32% |
|  | 4 | 3 | 0.91 | 0 | 0.00 | −0.95% |
| T | Unknown | 15 | 4.55 | 1 | 1.41 |  |
|  | 0 | 254 | 76.97 | 58 | 81.69 | 1.97% |
|  | 1 | 47 | 14.24 | 10 | 14.08 | −0.68% |
|  | 2 | 8 | 2.42 | 2 | 2.82 | 0.31% |
|  | 3 | 5 | 1.52 | 0 | 0.00 | −1.59% |
| N | Unknown | 16 | 4.85 | 1 | 1.41 |  |
| M | 0 | 307 | 93.03 | 68 | 95.77 | 0.97% |
|  | 1 | 3 | 0.91 | 0 | 0.00 | −0.97% |
|  | Unknown | 20 | 6.06 | 3 | 4.23 |  |
|  | In situ | 50 | 15.15 | 8 | 11.27 | −4.47% |
|  | I | 175 | 53.03 | 42 | 59.15 | 4.95% |
|  | II | 62 | 18.79 | 16 | 22.54 | 3.40% |
|  | III | 18 | 5.45 | 2 | 2.82 | −2.90% |
|  | IV | 3 | 0.91 | 0 | 0.00 | −0.97% |
| Clinical stage | Unknown | 22 | 6.67 | 3 | 4.23 |  |
| Total |  | 330 | 100.00 | 71 | 100.00 |  |

Pre-COVID totals calculated as the sum of the four pre-COVID rounds. Difference post–pre calculated comparing the post and the total pre-COVID percentages. Percentages show the distribution in the columns of each variable. p value of the distribution of each variable calculated with the exact test of Fisher. Dif = '^' indicates a significant difference of p > 0.05 between columns of the respective category.

any further testing after mammography. However, our data show that compliance with recall remained stable, indicating that women who chose to participate in the first place also engaged with further assessment when advised to do so.

Regarding the frequency of false positives, we found a statistically significant reduction in incident screening and no significant variation in prevalent screening. Considering that false positives and recall are closely related, these results are coherent if no changes in cancer detection rates were to be expected. Since recall and false positives are ideally supposed to be as low as possible, our results suggest that the diagnostic accuracy of the radiologists reading the mammograms was not materially affected by the pandemic. Similar pieces of evidence of the resilience of our public healthcare system have been recently reported in other hospitals in Barcelona (*Manzanares et al., 2021*), suggesting the strong resilience of health professionals working in critical situations. The COVID-19 pandemic has proved to be a stress test for healthcare systems around the world and the main elements related to highly effective responses have been associated with adaptation of health systems' capacity, reduction of vulnerability, preservation of healthcare functions and resources, and activation of comprehensive responses (*Haldane et al., 2021*).

We found no differences in the odds of screen-detected cancer for either prevalent or incident screenings when comparing the pre- and post-COVID-19 periods. In contrast to our statistical approach to estimate the cancer detection as the number of tumors per 1000 participants, when the absolute number of diagnoses during the interruption of the screening programs was compared with previous periods, an evident reduction was observed. A study performed in Málaga (Spain) reported that the breast was one of the cancer sites showing a larger decline in cases in April 2020 compared with April 2019. The authors of that study stated that this decline could be explained by the interruption of the screening program (*Ruiz-Medina et al., 2021*). Similar results have been found in studies from the Netherlands, Austria, and the United Kingdom (*Dinmohamed et al., 2020*; *Tsibulak et al., 2020*; *Limb, 2021*).

It is still unknown whether the target strategies to reduce the back-log of women who missed screening due to the pandemic, such as contacting them by telephone calls to schedule an appointment, will help to detect cancers missed during screening disruption. The possible influence of the delay on stage at diagnosis needs further evaluation. Although we found no statistically significant differences between pre- and post-COVID-19 periods in our small sample, we did find a small 5% increase in cases diagnosed at stage II. Similarly, an increased risk of late-stage breast cancer was observed in a month-by-month comparison in Israel in the period following the interruption and restoration of the screening activity (*Lloyd et al., 2021*). Further investigation on stage at diagnosis is essential, especially considering the potential increase in interval cancers due to the delay in the planned mammogram schedule. Moreover, the reduction in participation could increase cancer detection and stage at diagnosis in the next screening round among women who skipped the postpandemic round, leaving a span of 4 years between consecutive screenings.

Our study has some limitations. First, since some women could still have participated after the end of data collection, there could be a small bias overestimating the reduction in participation. Our experience shows that, except for very rare exceptions, women participate in the 3 months following the invitation, a period which was respected for all women in our study. Consequently, we believe the potential effect of this bias to be minimal for the results of our study. Second, the number of cancers detected during the pandemic period was relatively low, which limited the statistical power of our results and did not allow for multilevel models for this outcome. Nevertheless, we believe that, since cancer can only be detected once per woman, the nonindependence of the observations would not have a significant effect on this outcome.

Our study also has some strengths. To our best knowledge, most of the observational evidence assessing the effect of the pandemic compared screening indicators with the previous year (*Song et al., 2021*; *Chen et al., 2021*; *Toyoda et al., 2021*). In contrast, we included a long period of four previous rounds (8 years) of invitations for the same target population. We took this longitudinal approach since it is known that there are fluctuations in participation and cancer detection that may depend on time (*Giordano et al., 2015*). Therefore, our approach provides information on the pandemic beyond these common fluctuations. We also used a multilevel approach to our analyses in order to account for the nonindependence of the observations, and also provided stratified results according to each woman's previous history of participation.

In conclusion, our findings suggest that the impact of the pandemic on screening attendance depends on the type of screening, with women who regularly participate being the most affected. Targeting this specific population with a proactive invitation could be a way to ensure the historically higher participation in this group. Despite this, we should not forget other groups that attended screening less frequently. Our program has proved to be resilient, reducing recall and false positives while maintaining invitations and the cancer detection rate stable. These results suggest that the roll-out of the program was successful under the stressful situation provoked by the pandemic. Further prospective research is necessary to assess whether other factors played a role in participation during the pandemic, as well as to better characterize the impact of delays on stage at diagnosis and the incidence of interval cancers.

## Acknowledgements

We would like to thank Cristina Barrufet, Mercè Esturi, and Cristina Hernández in particular for their work and assistance in the performance of this study and the rest of the team of the PSMAR screening technical office: Isabel Amatriain, Gloria Lagarriga, Maria Ángeles Mercader, Marina Reyes, Judit Silvilla, and Eva Fernández.

## Additional information

### Funding

| Funder | Grant reference number | Author |
| --- | --- | --- |
| Instituto de Salud Carlos III | PI19/00007 | Margarita Posso<br>Xavier Castells |
| Instituto de Salud Carlos III | PI21/00058 | Marta Roman |

The funders had no role in study design, data collection, and interpretation, or the decision to submit the work for publication.

### Author contributions

Guillermo Bosch, Data curation, Formal analysis, Investigation, Methodology, Validation, Writing – original draft, Writing – review and editing; Margarita Posso, Conceptualization, Data curation, Formal analysis, Funding acquisition, Investigation, Methodology, Project administration, Supervision, Validation, Writing – original draft, Writing – review and editing; Javier Louro, Data curation, Formal analysis, Investigation, Methodology, Validation, Writing – review and editing; Marta Roman, Data curation, Formal analysis, Funding acquisition, Investigation, Methodology, Supervision, Writing – review and editing; Miquel Porta, Formal analysis, Investigation, Methodology, Supervision, Writing – review and editing; Xavier Castells, Conceptualization, Formal analysis, Funding acquisition, Investigation, Methodology, Supervision, Writing – review and editing; Francesc Macià, Conceptualization, Formal analysis, Investigation, Methodology, Project administration, Supervision, Validation, Writing – review and editing

### Author ORCIDs

Guillermo Bosch  http://orcid.org/0000-0001-5862-0913
Margarita Posso  http://orcid.org/0000-0002-5053-257X

### Ethics

Due to the retrospective nature of the study and the absence of direct contact with women, which did not affect their relationship with the program, informed consent was waived by the Ethics Committee of PSMAR, which approved the study (reg. 2021/9866). The study guaranteed Spain's legal regulations on data confidentiality (law 15/99 of December 13 on the protection of personal data).

### Decision letter and Author response

Decision letter https://doi.org/10.7554/eLife.77434.sa1
Author response https://doi.org/10.7554/eLife.77434.sa2

## Additional files

### Supplementary files
- Transparent reporting form
- Supplementary file 1. Crude logistic-regression models.

### Data availability
Source data form all tables and figures can be found in the following Dataset: BOSCH, GUILLERMO, 2022, "Breast cancer screening program invitations (2012-2021)", https://doi.org/10.7910/DVN/VVQNWM, Harvard Dataverse, V1, UNF:6:CaW3sEp4tMsg13z2I1eZbQ == [fileUNF] Data from "Impact on covid19 dataset invited women.sav" was used in tables 1 and 2 and figures 1,2 and 3. Data from "cancer characteristics database.tab" was used in table 3.

The following dataset was generated:

| Author(s) | Year | Dataset title | Dataset URL | Database and Identifier |
|---|---|---|---|---|
| Bosch G | 2022 | Breast cancer screening program invitations (2012-2021) | https://doi.org/10.7910/DVN/VVQNWM | Harvard Dataverse, 10.7910/DVN/VVQNWM |

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
