## [Editor Report]

This paper will be of interest to public health specialists and cancer scientists working in cancer prevention. The work presents valuable data on how the COVID-19 pandemic has impacted breast cancer screening indicators compared with previous years. Overall, the results support the assertion that while many key indicators have not been substantially impacted, the screening participation rate declined compared to the pre-pandemic era.

---

## [Decision Letter]

**Decision letter after peer review:**

Thank you for submitting your article "Impact of the COVID-19 pandemic on breast cancer screening indicators in a Spanish population-based program: a cohort study" for consideration by *eLife*. Your article has been reviewed by 3 peer reviewers, one of whom is a member of our Board of Reviewing Editors, and the evaluation has been overseen by a Senior Editor. The following individual involved in review of your submission has agreed to reveal their identity: Paolo Giorgi Rossi (Reviewer #2).

As is customary in *eLife*, the reviewers have discussed their critiques with one another. What follows below is the Reviewing Editor's edited compilation of the essential and ancillary points provided by reviewers in their critiques and in their interaction post-review. Please submit a revised version that addresses these concerns directly. Although we expect that you will address these comments in your response letter, we also need to see the corresponding revision clearly marked in the text of the manuscript. Some of the reviewers' comments may seem to be simple queries or challenges that do not prompt revisions to the text. Please keep in mind, however, that readers may have the same perspective as the reviewers. Therefore, it is essential that you attempt to amend or expand the text to clarify the narrative accordingly.

Essential revisions:

– Please provide an analysis of participation rates (participated/invited) additional to the distribution of participant distribution in Table 2 to adjust for changes in invitation distributions over time.

– Remove the description of the study as a "quasi-experimental" study.

– Please comment on the fact that recall rates and false positive rates are not independent outcomes.

– Consider performing a detailed monthly analysis of the selected indicators.

– Specify whether the analysis method accounted for the non-independence of the observations (multiple observations per woman). If not, please employ an analysis method that accounts for the correlation between observations, such as a multilevel model or a robust estimate of the variance.

– Specify the time frame considered for relevant outcomes such as participation in an invitation or detection.

– Please be careful not to interpret ORs as RRs. If the authors would like to use the results of the regression model as percentage reduction in the probability of an outcome, then they should use log-binomial models rather than logistic regression models.

– Please comment on whether changes in private care use may have had an impact on the results.

– Please comment on whether different screening invitation strategies were used during the pandemic compared with prior to the pandemic.

– If available, it would be interesting to also include participant compliance with recall for further assessment as an indicator to the analysis.

*Reviewer #1 (Recommendations for the authors):*

– Table 1 shows clearly that there were differences in the types of women who were invited in the pre- and post-COVID eras. This means that the differences in the types of participants in Table 2 reflect in part differences in invitations rather than differences in participation in the pre- and post-COVID era. Presumably, differences in invitation are due to changes in population composition/eligibility over time rather than changes attributable to the pandemic, and we would want to adjust for this when looking at whether participation has changed in the COVID era. Rather than examine participant distribution in Table 2, which is highly influenced by participant invitation, it would be more interesting to change Table 2 to look at participation rates instead (#participated/#invited) to see if the rates changed over time by participant characteristics.

– Discussion: "Although the aim of our study was not to evaluate the factors associated with participation, we found a lower representation of high-income women in the post-COVID-19 period." The lower proportion of high-income participants in the post-COVID-19 era is due to the lower proportion of high-income women which were invited in the post-COVID-19 era, and so may reflect changes in invitation rather than in participation. Please consider comparing the participation rates instead, as this will show whether high income women were less likely to participate in screening in the post-COVID-19 era than in the pre-COVID era.

– For Tables 1-3, the tests for significance appear to have been done by row rather for the whole RXC cross-tabulation distribution of a variable (for example, compare the % invitations in the 50-54 age group in pre-Covid to post-Covid in 2X2 tables, rather than compare the whole age distribution in pre-COVID vs post-COVID in the a RXC table). While this likely does not matter too much for Tables 1 & 2 due to the large number of participants, I think it could influence results in Table 3 due to the low number of cancer cases, which leads to lower statistical power when looking at individual rows. The authors should consider using the Freeman-Halton extension of the Fisher Test for contingency tables larger than 2X2 to compare the stage distributions across eras rather than comparing the 2X2 tables for individual cancer stages.

– Figure 2: I would suggest including the overall estimates as well as the stratified (prevalent/incident) estimates. It is possible the results for the recall would be significant if all events were combined together. The similarity of the ORs suggests that there is no interaction and that is justifiable to provide an overall estimate for all these outcomes.

– While the authors have posted their data and code in line with *eLife*'s data availability policy, the files are currently in restricted access. It would have been nice to be able to review these elements of the research as well.

– The analysis showed the cancer detection rate per number of participants was stable over time. However, I would have been interested also to see if the cancer detection rate per number of invitations has changed over time. Presumably, if participation is lower, then the overall yield of the program in terms of detection rate among all eligible (invited) women should be lower in the COVID-19 era.

*Reviewer #2 (Recommendations for the authors):*

I do not agree about the definition of a quasi-experimental study. I think this is a before/after study studying the impact of an external accident that cannot be considered as a natural experiment. The changes in screening activities that occurred as a consequence of the pandemic cannot be considered as a simulation of the consequences of a possible intervention (maybe suspending the invitations? shifting to opportunistic screening?). Thus, there is no possible intervention that we would like to evaluate in an experimental study design that could be assimilated to the pandemic. Furthermore, the pandemic had so many other impacts on society that, even if we could identify a theoretical similar intervention to our changes in the screening, the effects would be extremely confounded.

The interpretation of statistical significance in the case of recall and false-positive is tricky. The latter parameter should be fully determined given the recall rate and the detection rate. Therefore, if you are affirming that the recall rate increased in a way that is unlikely to be by chance, and detection is similar if not lower, the false positive cannot increase by chance but increased because the recall rate increased, even if the difference is not statistically significant.

Finally, a detailed analysis of the number of invitations per month or week and the mammographies performed could explain who actually delayed or decided not to attend at all. For example, did the women scheduled for the months of March and April attend less than those scheduled in September? In the previous attenders, can you compute the average delay since the previous examination month by month? This indicator would be very efficient in terms of statistical power and strictly related to the potential impact on health since it represents the actual delay in mammograms and potentially in diagnoses that occurred in women who attended… Given that the woman's id is the same through different rounds, you should be able to compute the distance between the previous mammogram ad present mammogram for three rounds before covid (n-1) and for covid round. the same could be done also for invitations; then delay in invitation could be used as a determinant of the participation. I think this would be a much more interesting analysis with more universal validity and relevance for the international audience.

Abstract

Methods: see general comments about "quasi-experimental".

please specify what do you mean with "observations"; invitations? screening episodes?

Results: I suggest not using the percentage difference in odds to describe the ORs (i.e. 11% lower) because this way of presenting results suggests that there was an 11% decrease in participation was, but actually the decrease is about 8% in relative terms and 3.7% as difference. I suggest using relative risks or prevalence ratios building log-binomial models, otherwise do not use 1-OR to report changes. Outcomes are very common and the OR is not a good estimator of the RR or the PR.

Introduction

It is well written, clear and focussed.

Methods

Page 4, line 10. Does the number refer to inhabitants or to the female target population? The second would be better.

Page 4 line 20: please explain better what you mean with "observations" invitation and I suppose all the consequent actions if the woman participated…

the sub-heading "outcomes" actually describes outcomes and co-variate or variables of interest.

Page 6 lines 6-9. this seems to be more relevant for analyses. did you really consider the observations regarding the same woman as independent? I suggest taking into account the possible non-independence of the observations using a multilevel model or a robust estimate of the variance. The effect should not be too important for stratified models or models with the interaction between screening round and Covid, but the estimate of the variance would be more adequate to the structure of the sample.

Page 6, line 20: participation in the round what does it mean? Which is the time since the invitation to be considered a participant? This is a critical point, in fact, if you included women who had the mammogram up to 20 months after the invitation, the follow-up time is shorter for the covid round than for the previous round. This could be a strong bias. Please explain what is the time considered for participating and demonstrate that there could not be a follow-up bias when comparing the last with previous rounds. This issue is also relevant for the detection rate.

Statistical analyses: see the previous comment about the structure of variance for non-independent observations.

Results

Page 8, line 12: please do not interpret 1-OR as 1-RR and thus a percentage of reduction. same in page 12 line 1.

In all the results I suggest reporting only one digit for decimals in raw percentages.

Table 3, probably even 1 digit for decimals for percentages is too much.

Discussion

The first sentence states "in this longitudinal study". I am not sure this study design is really a longitudinal study, i.e. a cohort. Actually, the study seems to be analysed as repeated cross-sectional surveys or as several short follow up cohorts (one for each round). I do not think it is really important to give a name or classify the study design, but in the methods, you did not present it as a cohort, so you should not introduce this concept in the discussion.

Page 17, lines 19-25: I suggest also another possible explanation, not mutually exclusive to the one proposed by the authors. Among women not attending screening, there are probably many women who perform mammography more or less regularly in opportunistic screening with private providers. During the lockdown, all the providers experienced difficulties in organizing planned activity as mammography in asymptomatic women. Therefore, it is possible that women who usually perform mammography in private had difficulties in getting their mammogram timely and when received the invitation by the program decided to participate. This could be a specular effect of that observed in women regularly attending the screening program that this time decided not to attend, probably because the invitation was late and they seek for mammography elsewhere. The disruption of planned activities during the lockdown, in all providers, may explain both decrease in participation in regular attendees and the increase in never attenders. This is also consistent with a decrease in high socioeconomic status in participants post covid, in fact, these women are those who may have a higher propensity to seek a mammogram in the private sector.

The authors correctly recognize the limit of assuming independence of the observation, but, if the id is woman-specific and is the same across the rounds, there is no need to make this assumption. I also agree that stratified analyses by type of round reduce the impact of this limit.

Page 19, line 1. "The same target population" this sentence may be confusing, the study does not have a closed cohort design, so each subsequent round targets slightly different populations.

*Reviewer #3 (Recommendations for the authors):*

I consider the work well done and interesting so for me is publishable with the following small revisions:

1. the authors describe what invitation modes were used up to the pandemic (invitation letter with prefixed appointment) but it is not clear what recruitment strategies were used during the pandemic. Were there any changes in recruitment methods? It would be useful for the readers to get an idea.

2. Page 4, lines 25-32; wouldn't this whole part go into the results?

3. One of the parameters used for the comparison is the recall rate. the authors limit their analysis the recall rate (number of women sent to the second level) and do not mention the compliance of women to in-depth diagnostics. Comparing the recall rates certainly allows to highlight a potential change in radiologists' behaviours, connected to all sanitization and social distancing measures that had to be put in place. It could have been interesting also to evaluate the presence (or not) of a change in women's behaviour and their willingness to compliance further assessments, despite the pandemic.

4. Among the analysed parameters, invitation coverage (the number of women invited out of the target population) was not included: it would have been interesting to know the trend of this parameter as an index of the resilience of the screening program.

5. the authors made a pre and post covid-19 pandemic comparison. It is a very marginal issue but I would not talk about post covid since we are still inside the pandemic (although hopefully in the final stages); what about pre covid and covid?

[Editors' note: further revisions were suggested prior to acceptance, as described below.]

Thank you for resubmitting your work entitled "Impact of the COVID-19 pandemic on breast cancer screening indicators in a Spanish population-based program: a cohort study" for further consideration by *eLife*. Your revised article has been evaluated by a Senior Editor and a Reviewing Editor.

The manuscript has been improved but there are some remaining issues that need to be addressed, as outlined below:

1. The first paragraph of the Discussion should be modified to reflect the new results, as some of the changes in recall and false positives are now significant.

2. The revision indicates that different invitation approaches were used for different types of participants (first, regular, non-participant) during the post-COVID era. Please discuss this in the Discussion, and how this may have impacted the results in Figure 1; it is likely some of the differences in participation may be due to these different invitation practices.

3. The analysis of compliance with recall (Table 5 in the response to reviewers) would be of interest to some readers and should be included in the manuscript.

---

## [Author Response]

Essential revisions:– Please provide an analysis of participation rates (participated/invited) additional to the distribution of participant distribution in Table 2 to adjust for changes in invitation distributions over time.

We thank the editor and reviewers for the comment and point of view on how to present the results of table 2. Although participation in our study is analyzed using regression models, we agree about the interest of presenting participations rates considering the changes in the invitation over time. Following this aim, we have added a new table to the manuscript (table 3) and we have modified the text in both the results and Discussion sections, to reflect the insight this new information provides. When we conceived table 2, we wanted to show that changes in invitation may translate into changes in the participant population. Therefore, due to the observed differences in the participants overtime, we have adjusted the analyses considering age, socio-economic status, and type of screening, to be able to isolate the effect of the pandemic on participation and the rest of the studied outcomes.

We have consequently modified the discussion in which our explanation may have been confusing, to clarify that even though the among all participants there were less who belonged to a high-income area, this was consequence of the changes in the invited population. The reduction in participation was greater in middle- and low-income areas, similarly to what other authors reported (1,2).

We have made the following changes to our manuscript:

Results (page 10):

“Analysis of participation proportions in different groups according to their characteristics revealed that participation decreased with age, with the largest reduction in participation occurring in women older than 65 years (-3.65%). Although participation decreased among all socioeconomic levels, the decrease was greater in those with middle-low (-6.03%) and low (-5.61%) status. Participation was greatly reduced amongst regular participants, who had participated in the previous round, with a 4.41% reduction, and slightly increased amongst previous non-participants, who participated for the first-time despite having been previously invited (+0.69%) (Table 3).”

Discussion (page 18)

“Overall, our findings on participation adjusted by age and socioeconomic status showed that the effect of the pandemic on screening attendance depended on each woman’s previous participation status. Although the aim of our study was not to evaluate the factors associated with participation, we found a lower representation of high-income women in the post-COVID-19 period, but this could probably be explained by the demographic changes in the invited population. Indeed, although participation decreased among all socio-economic groups, this decline was greater in low-income areas. A systematic review of studies conducted before the pandemic reported lower participation in low-income groups, immigrants, non-homeowners, and women with a previous false-positive result (3). Furthermore, studies recently published in the US have reported a decrease in participation, especially in underserved ethnic groups, with lower socioeconomic status, lack of insurance and longer travel time (1,2). Monitoring this information would allow programs to make efforts to promote participation among women at higher risk of not participating, especially under disruptive situations.”

Table 3 has also been updated to a new table containing this information, and the previous table 3 has changed its name to table 4.

– Remove the description of the study as a "quasi-experimental" study.

We thank the reviewers for their take on the description of our analysis and very much agree with the opinion of reviewer 2 that the changes in screening activities that occurred because of the pandemic cannot be considered as a simulation of the consequences of a possible intervention. Since the pandemic is an exposition over which we have no control, we have decided not to label our study as quasi-experimental, but as a before and after study, as which it was also referred to in other sections of the manuscript.

Changes have been made to the manuscript to reflect this, including the following:

Abstract (page 2)

“We conducted a before-and-after, study to evaluate participation, recall, false-positives, cancer detection rate, and cancer characteristics in our screening population from March 2020 to March 2021 compared with the four previous rounds (2012-2019). Using independent logistic regression models, we estimated the adjusted odds ratios (aOR) of each of the performance indicators for the COVID-19 period, controlling by type of screening (prevalent or incident), socioeconomic index, family history of breast cancer, and menopausal status. We analyzed 144,779 observations from 47,571 women.”

– Please comment on the fact that recall rates and false positive rates are not independent outcomes.

We agree with reviewer #2 in its interpretation of the correlation between recall rate and false positives, and we thank the interesting comment made on it. Since false positives depend on recall and cancer detection rates, in our study, with recall decreasing in the post-COVID period and cancer detection remaining stable, false positives should also decrease. With the changes in our statistic approach, we see how recall and false/positives aim in the same direction (decrease) after the pandemic, confirming the correlation of the outcomes.

We have modified our manuscript to comment on this observation.

Discussion (page 19)

“Despite the lower participation, the remaining performance indicators in our program did not seem to be negatively affected by the pandemic. Our results showed a statistically significant reduction in the recall rates of both prevalent and incident screening. These findings could be due to the increased workload caused by COVID-19 patients at our and many other hospitals, which strongly affected the radiology department in 2020 (4). Regarding the frequency of false positives, we found a statistically significant reduction in incident screening and no significant variation in prevalent screening. Considering that false-positives and recall are closely related, these results are coherent if no changes in cancer detection rates were to be expected. Since recall and false positives are ideally supposed to be as low as possible, our results suggest that the diagnostic accuracy of the radiologists reading the mammograms was not materially affected by the pandemic. Similar pieces of evidence of the resilience of our public health-care system have been recently reported in other hospitals in Barcelona (5), suggesting the strong resilience of health professionals working in critical situations. The COVID-19 pandemic has proved to be a stress test for health care systems around the world and the main elements related to highly effective responses have been associated with adaptation of health systems’ capacity, reduction of vulnerability, preservation of health care functions and resources, and activation of comprehensive responses (6).”

– Consider performing a detailed monthly analysis of the selected indicators.

We believe that, due to the organization of our program, a monthly analysis would be unprecise and would not reflect the real effect of the pandemic on the program. A monthly analysis could not show the impact on the main outcomes due to the fluctuations inherent to the program, since a specific chronogram of invitation is designed for each Basic Health Area based on its’ characteristics and population, ranging from 3 to 6 months each. Also, the small sample for each month would difficult the analyses and the delay in invitation would make for inaccurate comparisons between pre and post periods when preforming a monthly analysis.

We still think that it might be interesting to see how the number of participants and invitations fluctuated during the pre- and post-COVID periods. We designed the following graphs to illustrate these fluctuations and compare them between both periods. Since we only accounted for 10 out of the 25 Basic Health Areas included in our program, the number of invitations and mammograms are very uneven between odd and even years. For this reason, we accounted for the months in each biennial round, with months 1-12 representing even years and months 12-24 representing odd years. Since we wanted to assess the impact of the pandemic eruption during the following year, the Basic Health Areas selected were those for which invitations would have been sent out between March 2020 and March 2021. That is why in the las months of each round (from March of the second year) invitation and participation are almost 0, although some invitations can be issued out of the programmed period due to “manufactured” exceptions.

Author response image 1 shows how, for the post-covid screening round, the invitation process was suspended in April and May, with partial affectation of the months of march and June. From this point on, invitation and participation increased. The figure also shows how the logistical changes of the program affected the invitation process. Before the pandemic, no invitations were issued during the month of august (exceptions were made, which explains the toll higher than 0) but to regain the lost time, in august 2020 invitations were issued as done during the rest of the year, which explains the higher number invitations and mammograms, compared to the same month of previous years. The delay in invitations can also be perceived form the figure. The blue lines representing post-COVID period seem to follow the same tendencies than the green lines (pre-COVID mean) but later in time, which also causes for invitations to be issued in the post-COVID period until up to May 2021, when, for the selected BHAs, they should have ended by March like seen in the previous rounds. Higher peaks of invitation during the post-COVID period also reflect other logistical changes like the increased mammographic capacity obtained by increasing opening hours during weekdays and offering appointments on Saturdays.

**Author response image 1. sa2fig1:** 

– Specify whether the analysis method accounted for the non-independence of the observations (multiple observations per woman). If not, please employ an analysis method that accounts for the correlation between observations, such as a multilevel model or a robust estimate of the variance.

We initially intended not to account for the non-independence of the observations due to the complexity of the analyses. Since we were aware of the relationship between previous participations and the probability of participation for each woman, we stratified our analyses by type of screening round, which could mitigate, at least partially, the non-independence of the observations that he had not considered when choosing the analytic approach.

Following the recommendations issued by the editors and reviewers we have proceeded to change our analysis’ approach, using multi-level models for our logistic regressions in order to account for the non-independence of the same woman’s participations. Following this new approach, we have obtained an aOR of 0.90 (95%CI 0.84-0.96) for participation in the post-COVID-19 round for first-time invitees. For regular participants the aOR for participation was 0.63 (95%CI 0.59-0.67) and for irregular participants, aOR = 0.95 (95%CI 0.86-1.05). For previous non-participants the aOR = 1.10 (95%CI 1.01-1.20).

These results are very similar to the ones previously obtained with the independent logistic regression models, showing our approach was a good approximation to the multi-level models. The only remarkable change is the statistical significance of the increased odds of participation in previous non-participants, although the punctual estimate has barely changed (from aOR = 1.07 to aOR=1.10).

The same analytic approach has been used for recall and false-positive results. The aOR of recall between the post and pre-COVID-19 periods was 0.74 (95%CI 0.56-0.99) for prevalent screening and aOR = 0.80 (95%CI 0.68-0.95) for incident screening. The aOR of false positives was 0.92 (95%CI 0.66-1.28) for prevalent screening, and 0.72 (95%CI 0.59-0.88) for incident screening.

Compared to our previous approach, recall was lower using the multi-leveled analysis and this difference between the re and post-COVID-19 periods became statistically significant. These results aim in the same direction as we had previously stated so our discussion is still valid and even more relevant as the reduction in recall seems to be bigger than previously found. For false positives the multilevel approach changes the direction of our results for prevalent screening showing a reduction in false-positives for the post-COVID-19 period. This result is much more in line to what we wound have expected given the reduction in recall and the stability in cancer detection, as further discussed in a previous section of this letter.

For cancer detection we would argue that our previous approach with independent measurements is still valid, since cancer can be only detected once in the screening program for a same woman, and no repeated measurements can be obtained. Even though we tried to use the same multi-level analyses as in the previous outcomes, our models failed to converge. Still, we believe the non-independence of the measurements should not have an effect in the effect of the pandemic over cancer detection.

We have proceeded to incorporate the following changes in our manuscript to reflect the new analytic approach and its consequences on the results of our study.

Abstract (page 2)

Methods:

“We conducted a before-and-after, study to evaluate participation, recall, false-positives, cancer detection rate, and cancer characteristics in our screening population from March 2020 to March 2021 compared with the four previous rounds (2012-2019). Using multi-level logistic regression models, we estimated the adjusted odds ratios (aOR) of each of the performance indicators for the COVID-19 period, controlling by type of screening (prevalent or incident), socioeconomic index, family history of breast cancer, and menopausal status. We analyzed 144,779 invitations from 47,571 women.”

Results

“During the COVID-19 period, the odds of participation were lower in first-time invitees (aOR=0.90[95%CI=0.84-0.96]) and in those who had previously participated regularly and irregularly (aOR=0.63 [95%CI=0.59-0.67] and aOR=0.95 [95%CI=0.86-1.05], respectively). Participation showed a modest increase in women not attending any of the previous rounds (aOR=1.10 [95%CI=1.01-1.20]). The recall rate decreased in both prevalent and incident screening (aOR=0.74 [95%CI 0.56-0.99] and aOR=0.80 [95%CI 0.68-0.95] , respectively). False positives also decreased for both groups (prevalent aOR=0.92 [95%CI 0.66-1.28] and incident aOR=0.72 [95%CI 0.59-0.88]). No significant differences were observed in cancer detection rate (aOR=0.91 [95%CI=0.69-1.18]), or cancer stages.”

Conclusions:

“The COVID-19 pandemic negatively affected screening attendance, especially in previous participants and newcomers. We found a reduction in recall, false-positives, and no marked differences for cancer detection, indicating the program’s resilience. There is a need for further evaluations of interval cancers and potential diagnostic delays.”

Methods (page 7)

“We first compared the characteristics of the invited population among the different screening rounds to describe variations in their distribution. We evaluated differences in the categories using the chi-square test or the exact Fisher test when appropriate.

Then, we created multilevel logistic regression models to estimate adjusted odds ratios (aOR) of each of the performance indicators and their corresponding 95% confidence intervals (95% CI) for the COVID-19 period, adjusting by the clinically relevant variables.

For participation, we included the following variables in the model: type of screening round (prevalent vs. incident), age group, and socio-economic index. We found a strong interaction between COVID-19 and the type of screening round. Therefore, we created a new variable, which represented this interaction. Hence, the final models for participation differentiated four screening groups (prevalent-first-time invitee, prevalent-previous nonparticipant, incident-regular participant, and incident-irregular participant). We obtained crude results and adjusted by age and socioeconomic index.

We created three additional models, including only participants, to assess the impact of COVID-19 on the other main indicators of the screening program: recalland false-positives. Finally, we used independent logistic regression models for the screen-detected cancer rate (invasive or in situ). These models were adjusted for age group, menopausal status, and breast cancer family history.

Finally, we compared the stage at diagnosis and the remaining cancer characteristics (size, lymph node invasion, and metastasis invasion) of cases detected in the screening program in the pre- and post-COVID-19 periods.”

Results (page 8)

“Participation in the program was affected differently depending on the type of screening. The aOR of participation between the post and pre-COVID-19 periods was 0.90 (95% CI, 0.84-0.96) for the group of first-time invitees. The aOR was 1.10 (95% CI, 1.01-1.20) for the previous non-participant group between the post- and pre-COVID-19 periods. For the group of women who had participated in the previous round (regular participants), the aOR of participation was 0.63 (95% CI 0.59-0.67), and for those not participating in the last round (irregular participants), the aOR was 0.95 (95% CI 0.86-1.05) (Figure 1).”

Results (page 13)

“Analysis of recall revealed modest decreases in the odds of being advised to undergo additional testing during the post-COVID-19 period in both the prevalent and the incident screening groups (aOR=0.74 [95%CI 0.56-0.99] and aOR=0.80 [95%CI 0.68-0.95] ). The aOR of a false positive result for prevalent and incident screening was 0.92 (95%CI 0.66-1.28) and 0.72 (95%CI 0.59-0.88) , respectively. The aOR of cancer detection in the post-COVID vs the pre-COVID-19 period was 1.01 (95% CI 0.56-1.71) and 0.87 (95% CI 0.63-1.17) in the prevalent and incident screening groups, respectively (Figure 2).”

Discussion (page 18)

“Women who had been previously invited but had never attended our screening invitation seemed to participate slightly more during the pandemic period. The increase in participation was not expected since this group of women is that with the lowest participation in our setting (7). However, this change could be explained by a plausible modification in attitudes to screening with a possible increase in health-consciousness promoted by the pandemic, prompting women who had never been interested in screening to participate for the first time. Women who have never previously participated due to private screening may also have switched to the population-based program due to the effect of the pandemic on private clinics, which also had to stop their preventive care programs during lockdown.”

Discussion (page 19)

“Despite the lower participation, the remaining performance indicators in our program did not seem to be negatively affected by the pandemic. Our results showed a statistically significant reduction in the recall rates of both prevalent and incident screening. These findings could be due to the increased workload caused by COVID-19 patients at our and many other hospitals, which strongly affected the radiology department in 2020 (4). Regarding the frequency of false positives, we found a statistically significant reduction in incident screening and no significant variation in prevalent screening. Considering that false-positives and recall are closely related, these results are coherent if no changes in cancer detection rates were to be expected.”

Discussion (page 19)

“Our study has some limitations. First, since some women could still have participated after the end of data collection, there could be a small bias overestimating the reduction in participation. Our experience shows that, except for very rare exceptions, women participate in the 3 months following the invitation, a period which was respected for all women in our study. Consequently, we believe the potential effect of this bias to be minimal for the results of our study. Second, the number of cancers detected during the pandemic period was relatively low, which limited the statistical power of our results and did not allow for multi-level models for this outcome. Nevertheless, we believe that, since cancer can only be detected once per woman, the non-independence of the observations would not have a significant effect on this outcome.”

Discussion (page 20)

“Our study also has some strengths. To our best knowledge, most of the observational evidence assessing the effect of the pandemic compared screening indicators with the previous year (8–10). In contrast, we included a long period of four previous rounds (eight years) of invitations for the same target population. We took this longitudinal approach since it is known that there are fluctuations in participation and cancer detection that may depend on time (11). Therefore, our approach provides information on the pandemic beyond these common fluctuations. We also used a multi-level approach to our analyses in order to account for the non-independence of the observations, as well as offering stratified results according to the previous history of participation of each women.”

Discussion (page 20)

“Our program has proved to be resilient, reducing recall and false positives while maintaining invitations and the cancer detection rate stable. These results suggest that the roll-out of the program was successful under the stressful situation provoked by the pandemic.”

– Specify the time frame considered for relevant outcomes such as participation in an invitation or detection.

There was not a specific time frame considered for each outcome. We selected the women who should have been invited from March 2020 to March 2021. Since our invitation process follows geographical criteria and we don’t have data on the individual date for when a woman should have been invited, we selected the areas to which invitation should have been sent out from March 2020 to March 2021. Some of these areas had beginning sending out invitations before March 2020 and, due to the interruption and delay in the program, some areas were sending invitations later than March 2021, although all invitations for the selected areas had been sent out by the time data was collected (October 2021). In our program, almost all participation happens in the month following invitation, but rare cases could happen up to 6 months prior to the next invitation. So, for our post-covid round some women could still have participated after data collection, but it is hard to that this exceptional out-of-calendar participation could explain the differences found in our study between pre- and post-COVID periods’ participation.

The rest of the outcomes are not time-dependent since they are linked to the screening episode. This way, when a cancer is detected following screening, it is linked to the screening invitation it followed. Although almost all invitations had been sent by June 2021, we collected data up to October 2021 to account for the delay between invitation and cancer detection.

– Please be careful not to interpret ORs as RRs. If the authors would like to use the results of the regression model as percentage reduction in the probability of an outcome, then they should use log-binomial models rather than logistic regression models.

We are thankful for the observation on the interpretation and OR and agree that our redaction could be clearer in order to avoid misunderstandings on the magnitude of our results. Consequently, we have proceeded to modify our manuscript as follows:

Abstract (page 2)

“During the COVID-19 period, the odds of participation were lower in first-time invitees (aOR=0.90[95%CI=0.84-0.96]) and in those who had previously participated regularly and irregularly (aOR=0.63 [95%CI=0.59-0.67] and aOR=0.95 [95%CI=0.86-1.05], respectively). Participation showed a modest increase in women not attending any of the previous rounds (aOR=1.10 [95%CI=1.01-1.20]). The recall rate decreased in both prevalent and incident screening (aOR=0.74 [95%CI 0.56-0.99] and aOR=0.80 [95%CI 0.68-0.95] , respectively). False positives also decreased for both groups (prevalent aOR=0.92 [95%CI 0.66-1.28] and incident aOR=0.72 [95%CI 0.59-0.88]). No significant differences were observed in cancer detection rate (aOR=0.91 [95%CI=0.69-1.18]), or cancer stages.”

Results (page 8)

“Participation in the program was affected differently depending on the type of screening. The aOR of participation between the post- and pre-COVID-19 periods was 0.90 (95% CI, 0.84-0.96) for the group of first-time invitees. The aOR was 1.10 (95% CI, 1.01-1.20) for the previous non-participant group between the post- and pre-COVID-19 periods. For the group of women who had participated in the previous round (regular participants), the aOR of participation was 0.63 (95% CI 0.59-0.67), and for those not participating in the last round (irregular participants), the aOR was 0.95 (95% CI 0.86-1.05) (Figure 1).”

Results (page 13)

“Analysis of recall revealed modest decreases in the odds of being advised to undergo additional testing during the post-COVID-19 period in both the prevalent and the incident screening groups (aOR=0.74 [95%CI 0.56-0.99] and aOR=0.80 [95%CI 0.68-0.95] ). The aOR of a false positive result for prevalent and incident screening was 0.92 (95%CI 0.66-1.28) and 0.72 (95%CI 0.59-0.88), respectively. The aOR of cancer detection in the post-COVID vs the pre-COVID-19 period was 1.01 (95% CI 0.56-1.71) and 0.87 (95% CI 0.63-1.17) in the prevalent and incident screening groups, respectively (Figure 2).”

– Please comment on whether changes in private care use may have had an impact on the results.

The observation made by reviewer #3 on the role of private care seems very relevant to us, and we thank the comment on the matter. Private-care screening is very common in Catalonia, which makes this a very important matter that we must account for whenever we are managing data from our program, in relation to its population-related impact. We generally hypothesize that private screening accounts for a low-participation in the population-based program, but we estimate that around 80% of invited women are screened when combining population-based and privately provided screenings. When designing our study, we hypothesized that women with private health-care providers might have preferred to be screened elsewhere to avoid the possible risk related to bigger public hospitals. We see how the lower percentage of participation among higher socio-economic status areas may confirm this theory. Despite this, it is also plausible that private health providers, with less resources than public-funded hospitals, were also overwhelmed by the workload of the pandemic, delaying or cancelling non-urgent procedures. Women who do not participate in the population-based program may have received the invitation before they were able to attend their private-care provider and might have decide to participate in the population-based program for the first time. This behavior may also explain the increased participation in previous non-participants seen in our logistic regression model.

We have modified the manuscript to reflect this reasoning as shown below:

-Discussion (page 18):

“Women who had been previously invited but had never attended our screening invitation seemed to participate slightly more during the pandemic period. The increase in participation was not expected since this group of women is that with the lowest participation in our setting (7). However, this change could be explained by a plausible modification in attitudes to screening with a possible increase in health-consciousness promoted by the pandemic, prompting women who had never been interested in screening to participate for the first time. Women who have never previously participated due to private screening may also have switched to the population-based program due to the effect of the pandemic on private clinics, which also had to stop their preventive care programs during lockdown.”

– Please comment on whether different screening invitation strategies were used during the pandemic compared with prior to the pandemic.

With the intention of reducing the extension of our manuscript, only a brief resume of the invitation process was included. We would like to provide the reviewers with additional information on how the invitation process was being managed before the pandemic-induced interruption, and how it has changed.

At the beginning of each screening round information on the new eligible women is obtained for the Central Registry of Public Insurance (RCA) and added to the existing program management database. Invitations are automatically issued for eligible women if they have not been permanently excluded by any exclusion criteria. Each woman is given a mammogram appointment and a letter of invitation is issued and sent to his home address. Women are asked to call to a specific phone number if they wish to change the date of the mammogram. If a woman does not show up to her mammography appointment a second letter is sent to her, offering a new mammography appointment. In case of not participating, she gets a phone call from the program's staff to remind her and either set a new appointment or record the motive of non-participation. This episode remains open until six months prior to the beginning of the next screening round. A woman can call for an appointment anytime if she changes her mind, and if she doesn’t, she is then appointed as non-participant.

If the woman shows up to the first invitation appointment, her episode will be closed once the mammography test is read as negative, or a diagnosis is confirmed. In both cases the woman is counted as participant. If a cancer is detected, she will be excluded of subsequent screenings and enter the therapeutic circuits.

The delay caused by the three months of interruption of the screening program due to the first wave of the pandemic, and the lower volume of mammograms per hour due to preventive measures to reduce waiting times in the hospital, made it necessary for some organizational changes.

In the first place, mammography opening hours increased, with an extra hour on the evenings and appointment on Saturday mornings, allowing for more mammograms to be scheduled every day despite the preventive measures.

On the other hand, some changes were made in the invitation process. Regular participants, being the group with the highest chance of participating, were called by telephone to set the mammography test, instead of giving them a preset appointment by postal mail. Nevertheless, a previous letter was sent to them to inform them about the upcoming call. The same process was used for women invited for the first time. Previous non-participants were invited by postal mail as usual but, instead of having a pre-set date, they were asked to call to set an appointment if they wished to participate. Women who didn’t call asking for an appointment, predicting a high absence, were sent an invitation for a preset appointment, in specific dates programmed with overbooking. This way, absences could be reduced and mammographers were used more efficiently, while still setting mammography appointment for all invited women.

The manuscript has been modified in order to introduce addition information on the invitation process after the pandemic, while trying to be as concise as possible:

Methods (page 4):

“Until the pandemic, screening invitations were sent by postal mail with a pre-scheduled mammogram appointment to all invited women. Since 2020, previous participants and first-time invitees have been invited to participate by telephone, in addition to a previously sent letter informing them of the upcoming call. Previous non-participants still receive an invitation via postal mail, without a pre-set date, inviting them to call a specific telephone number to schedule the mammogram at a convenient time.”

– If available, it would be interesting to also include participant compliance with recall for further assessment as an indicator to the analysis.

We thank the editors and reviewer #3 for their thoughts on the addition to compliance with recall as one of our indicators. It is a very interesting hypothesis to suggest that diagnostic confirmation procedures’ attendance could decrease due to the COVID-19 induced situation. Although all the diagnostic circuits were active during and after the most critical months of the pandemic, it is possible that women would decline going through additional testing due to fear of the hospital’s safety our due to other more prominent health concerns, specially COVID-19 infection. Unfortunately, our small sample of women recalled for further assessment makes it very inconvenient to estimate the compliance to recall. We have obtained information on the result of the additional diagnostic testing preformed in women recalled to our program and we have assumed that women with no result didn’t comply with further assessments. This is not necessarily true since these women could have chosen to seek further assessment in some other medical center through private health coverage, as we have already comment on with previous steeps of the screening process.

Following this approach as an approximation we have obtained the data shown in Table 5.

As shown in table 5, compliance with recall is very stable and has varied a maximum of 3 percentual points in the last years. Although it was a little lower in the post-COVID round than in the previous one, it was even lower in the first screening round studied (corresponding to the years 2012-2015). The small change in tendency could be due to the pandemic effect but it is not as clear as seen with other outcomes like participation and could be due to intrinsic fluctuations. We used a simple unadjusted logistic regression model to compare the odds of adherence to recall before and after the pandemic which showed a discreet non-statistically significant increase (OR = 1.26, 95%CI 0.76-2.23). We could not adjust according to screening history or any other covariant due to the small number of cases.

The increased odds of compliance to recall, even if not statistically significant, seems to aim against our initial hypothesis but this could be to the great decrease seen over the last couple of rounds. It could also be indicative of a real effect, since it is possible that women who participated during the post-covid period were more health-conscious and having made the effort to participate in screening in the first place, where keener to undergo further assessment if recommended.

We feel these results are not especially relevant to our study and have decided to include it into the manuscript, to focus on our main results and keep it as simple and comprehensive as possible, given the complexity of the matter at study. Nevertheless, if the editors find these results to be relevant to the readers, they could be incorporated to the manuscript.

Reviewer #1 (Recommendations for the authors):– Table 1 shows clearly that there were differences in the types of women who were invited in the pre- and post-COVID eras. This means that the differences in the types of participants in Table 2 reflect in part differences in invitations rather than differences in participation in the pre- and post-COVID era. Presumably, differences in invitation are due to changes in population composition/eligibility over time rather than changes attributable to the pandemic, and we would want to adjust for this when looking at whether participation has changed in the COVID era. Rather than examine participant distribution in Table 2, which is highly influenced by participant invitation, it would be more interesting to change Table 2 to look at participation rates instead (#participated/#invited) to see if the rates changed over time by participant characteristics.

We thank the reviewer for the insight and have already addressed this issue in the essential revisions section of this letter. As recommended by the editor, we have added a complementary table to table 2, table 3, which shows the participation rate for each one of the categories for the variables studied. We have reflected our finding on both the results and Discussion sections of the manuscript. Nevertheless, these changes do not affect the validity of our modeled results since all models were adjusted for age and socio-economic status.

– Discussion: "Although the aim of our study was not to evaluate the factors associated with participation, we found a lower representation of high-income women in the post-COVID-19 period." The lower proportion of high-income participants in the post-COVID-19 era is due to the lower proportion of high-income women which were invited in the post-COVID-19 era, and so may reflect changes in invitation rather than in participation. Please consider comparing the participation rates instead, as this will show whether high income women were less likely to participate in screening in the post-COVID-19 era than in the pre-COVID era.

This comment has also been addressed in the essential revisions section, along the previous one since they are closely related. As shown in new table 3, participation decreased amongst all socio-economic groups, but the decrease was higher for lower socio-economic status. The manuscript has been updated as shown in the essential revisions section of these letter, to clarify our findings.

– For Tables 1-3, the tests for significance appear to have been done by row rather for the whole RXC cross-tabulation distribution of a variable (for example, compare the % invitations in the 50-54 age group in pre-Covid to post-Covid in 2X2 tables, rather than compare the whole age distribution in pre-COVID vs post-COVID in the a RXC table). While this likely does not matter too much for Tables 1 & 2 due to the large number of participants, I think it could influence results in Table 3 due to the low number of cancer cases, which leads to lower statistical power when looking at individual rows. The authors should consider using the Freeman-Halton extension of the Fisher Test for contingency tables larger than 2X2 to compare the stage distributions across eras rather than comparing the 2X2 tables for individual cancer stages.

We thank the reviewer for the insight on this matter and agree on the alternative method for comparing the different eras in table 1-3. We did consider this approach in previous stages of our study but finally decided to compare each category separately. We felt that, since we provided the percentual difference for each category, it could be confusing to compare the whole distribution of each variable. It is true that, for table 3, this could have been useful given the small number of cases, but we preferred to maintain the same approach as in the other tables, to allow an easier interpretation. Nevertheless, we would like to inform the reviewers that no p values < 0.05 were found for table 3 when using the Freeman-Halton distribution, so it provides no additional information to the one shown in the table.

– Figure 2: I would suggest including the overall estimates as well as the stratified (prevalent/incident) estimates. It is possible the results for the recall would be significant if all events were combined together. The similarity of the ORs suggests that there is no interaction and that is justifiable to provide an overall estimate for all these outcomes.

For outcomes other than participation, we decided to differentiate the estimates between prevalent and incident screening due to the big difference in the incidence of these indicators in the first screening compared to the rest. In figures 2 and 3, we show not only the OR but also the incidence difference between both periods and we felt this big difference was relevant enough to be shown, even if the aOR are alike for both groups. Only for cancers, we considered it to be necessary to show an overall estimate (figure 3), due to the small number of cases.

Nevertheless, we did create additional models to obtain overall estimates for both recall and false positives. These models generated very similar estimates to the ones obtained when stratified according to type of screening. The aOR for overall recall was 0.71 (95%CI 0.61-0.82). and for overall false positive the aOR was 0.67 (95%CI 0.56-0.79). We feel these results do not add any relevant data to the ones shown in figure 2. In fact, they are very similar to those seen for incident screening, since most of the participants belong to this group. With the aim of simplifying the presentation of our findings have chosen not to include them in the manuscript.

– While the authors have posted their data and code in line with eLife's data availability policy, the files are currently in restricted access. It would have been nice to be able to review these elements of the research as well.

We would like to apologize to the reviewer for any complication derived from the data repository access. We have granted access as swiftly as possible to everyone who has solicited it.

– The analysis showed the cancer detection rate per number of participants was stable over time. However, I would have been interested also to see if the cancer detection rate per number of invitations has changed over time. Presumably, if participation is lower, then the overall yield of the program in terms of detection rate among all eligible (invited) women should be lower in the COVID-19 era.

We thank the reviewer for the interest shown on this outcome. Nevertheless, the denominator for cancer detection rates in our program is always the number of participants. We agree on the relevance of the outcome proposed by the reviewer, but we believe that, in order to simplify or results, it is clearer to keep the most wildly used denominator for cancer detection rates. Considering that the number of invitations has remained relatively stable over the years, an approximation to the outcome proposed by the reviewer can be obtained from table 3, where the total number of cancers for the pre- and post-COVID-19 periods is informed.

Reviewer #2 (Recommendations for the authors):General comments:I do not agree about the definition of a quasi-experimental study. I think this is a before/after study studying the impact of an external accident that cannot be considered as a natural experiment. The changes in screening activities that occurred as a consequence of the pandemic cannot be considered as a simulation of the consequences of a possible intervention (maybe suspending the invitations? shifting to opportunistic screening?). Thus, there is no possible intervention that we would like to evaluate in an experimental study design that could be assimilated to the pandemic. Furthermore, the pandemic had so many other impacts on society that, even if we could identify a theoretical similar intervention to our changes in the screening, the effects would be extremely confounded.

We thank the reviewer for the comment and agree with the reasoning exposed. This topic has already been addressed in the essential revisions section of this letter, and the manuscript has been corrected accordingly.

The interpretation of statistical significance in the case of recall and false-positive is tricky. The latter parameter should be fully determined given the recall rate and the detection rate. Therefore, if you are affirming that the recall rate increased in a way that is unlikely to be by chance, and detection is similar if not lower, the false positive cannot increase by chance but increased because the recall rate increased, even if the difference is not statistically significant.

We thank the reviewer for the very interesting comment on the relation between recall and false positives. It has helped us to further reflect on this correlation. This topic has also been already addressed in the essential revision section of this letter.

Finally, a detailed analysis of the number of invitations per month or week and the mammographies performed could explain who actually delayed or decided not to attend at all. For example, did the women scheduled for the months of March and April attend less than those scheduled in September? In the previous attenders, can you compute the average delay since the previous examination month by month? This indicator would be very efficient in terms of statistical power and strictly related to the potential impact on health since it represents the actual delay in mammograms and potentially in diagnoses that occurred in women who attended… Given that the woman's id is the same through different rounds, you should be able to compute the distance between the previous mammogram ad present mammogram for three rounds before covid (n-1) and for covid round. the same could be done also for invitations; then delay in invitation could be used as a determinant of the participation. I think this would be a much more interesting analysis with more universal validity and relevance for the international audience.

We thank the reviewer’s suggestions on the analysis approach. Unfortunately, our program’s organization hinders such monthly analyses. This topic has been addressed in the essential revisions, where approximations to the monthly analyses of participation are further discussed.

Additionally, we would like to note that our manuscript is the first publication of a publicly funded project which aims to further investigate the effects of the pandemic on the breast cancer screening program looking further into other outcomes like interval cancers. We believe some of the questions and analyses suggested by the reviewer will be addressed by the following publications. Nevertheless, we have performed a preliminary analysis on the changes in time between successive invitations.

In Author response image 2 we see how the time between invitations increased in more than 4 months after the interruption caused by the COVID-19 lockdown. We only accounted for 10 Basic Health Areas in our study, so information beyond month 16 (April of the second year for each round) should not be considered for the interpretation of the figure.

AbstractMethods: see general comments about "quasi-experimental".

This topic has already been addressed in the essential submissions section.

Please specify what do you mean with "observations"; invitations? screening episodes?

The term “observation” is used in our manuscript to refer to every screening episode, which account for an invitation to screening any further testing following that invitation (participation, recall, cancer detection).

In the abstract’s methods section, we have changed the term observations for “invitations” to avoid possible misunderstandings as shown below:

Abstracts (page 2):

“We conducted a before-and-after, study to evaluate participation, recall, false-positives, cancer detection rate, and cancer characteristics in our screening population from March 2020 to March 2021 compared with the four previous rounds (2012-2019). Using independent logistic regression models, we estimated the adjusted odds ratios (aOR) of each of the performance indicators for the COVID-19 period, controlling by type of screening (prevalent or incident), socioeconomic index, family history of breast cancer, and menopausal status. We analyzed 144,779 invitations from 47,571 women.”

Results: I suggest not using the percentage difference in odds to describe the ORs (i.e. 11% lower) because this way of presenting results suggests that there was an 11% decrease in participation was, but actually the decrease is about 8% in relative terms and 3.7% as difference. I suggest using relative risks or prevalence ratios building log-binomial models, otherwise do not use 1-OR to report changes. Outcomes are very common and the OR is not a good estimator of the RR or the PR.

We thank the reviewer for the suggestions, and it has already been addressed in the essential revisions section.

IntroductionIt is well written, clear and focussed.MethodsPage 4, line 10. Does the number refer to inhabitants or to the female target population? The second would be better.

The number shown in the previous version of the manuscript refers to the total population. The manuscript has been updated to clarify this matter and include the approximate number of eligible women.

Methods (page 4)

“In Spain, publicly funded mammographic screening for breast cancer is offered every two years to women aged 50 to 69 years (12). The screening examination at PSMAR consists of both a mediolateral oblique and a craniocaudal digital (two-dimensional) mammographic view of each breast. Two independent radiologists with extensive experience perform blinded double reading of mammograms. Disagreements are resolved by a third senior radiologist (13). The program covers the population of four districts of the city of Barcelona, with around 620,000 inhabitants, and approximately 75,000 eligible women.”

Page 4 line 20: please explain better what you mean with "observations" invitation and I suppose all the consequent actions if the woman participated…

As previously addressed, the term “observation” is used in our manuscript to refer to every screening episode, which account for an invitation to screening any further testing following that invitation (participation, recall, cancer detection).

We have corrected the methods section of the manuscript to explain the meaning of this term better. The paragraph has been modified as shown below:

Methods (page 4):

“We obtained a total of 144,779 observations, which are screening invitations linked with the following actions that may follow them (participation, recall, cancer detection), from 47,571 eligible women throughout the 10 years of study. Each of these observations represented an invitation to the screening program. In our study population, age group, socioeconomic status, and type of screening round were statistically different in the post- and pre-COVID-19 periods, (Table 1). The percentage of invited women living in high-income areas decreased slightly (-1.03%) as did that of women younger than 55 years (-1.83%). The distribution of the type of screening of invited women also changed, with a higher percentage of invitations for prevalent screening (+1.69%), especially first-time invitees (+2.90%).”

the sub-heading "outcomes" actually describes outcomes and co-variate or variables of interest.

We agree with the reviewer’s observation and have decided to modify the sub-heading to better suit the content of the section.

We have made the following changes to sub-heading in the manuscript:

“Outcomes and variables of interest”

Page 6 lines 6-9. this seems to be more relevant for analyses. did you really consider the observations regarding the same woman as independent? I suggest taking into account the possible non-independence of the observations using a multilevel model or a robust estimate of the variance. The effect should not be too important for stratified models or models with the interaction between screening round and Covid, but the estimate of the variance would be more adequate to the structure of the sample.

We thank the reviewer’s comments and have addressed this issue in the essential revisions section of this letter.

Page 6, line 20: participation in the round what does it mean? Which is the time since the invitation to be considered a participant? This is a critical point, in fact, if you included women who had the mammogram up to 20 months after the invitation, the follow-up time is shorter for the covid round than for the previous round. This could be a strong bias. Please explain what is the time considered for participating and demonstrate that there could not be a follow-up bias when comparing the last with previous rounds. This issue is also relevant for the detection rate.Statistical analyses: see the previous comment about the structure of variance for non-independent observations.

We thank the reviewer’s comment on this issue, since it was one of the limitations we had to manage during the study design. As already addressed in the essential revisions section of this letter, a woman can choose to participate up to 6 months prior to its next invitation, which means participation can happen in the 18 months following invitation. Nevertheless, our expertise tells us that almost all participants take their mammogram in the month following invitation, or even up to 3 months after. Preset appointments are usually programmed 2 weeks after sending the invitation letter. The supplementary figure 1 that clearly shows how participation beyond this timeframe is very infrequent. In addition to this, we have to consider the changes in the invitations process implemented due to the pandemic and which are also explained into detail in previous sections of this letter. Since previous participants make for the larger group of participants and they were contacted by phone to make an appointment, these appointments were probably closer to the invitation than in previous screening rounds. We believe the potential follow-up bias could be very much addressed by these two circumstances but should of course be considered as a limitation, although it is very unlikely that it can explain the results of our study.

We have updated our manuscript in order to better explain this limitation as follows:

Discussion (page 19):

“Our study has some limitations. First, since some women could still have participated after the end of data collection, there could be a small bias overestimating the reduction in participation. Our experience shows that, except for very rare exceptions, women participate in the 3 months following the invitation, a period which was respected for all women in our study. Consequently, we believe the potential effect of this bias to be minimal for the results of our study.”

ResultsPage 8, line 12: please do not interpret 1-OR as 1-RR and thus a percentage of reduction. same in page 12 line 1.

We thank the reviewer’s comment and have already change our manuscript accordingly. We have already addressed these changes in the essential revisions section of this letter.

In all the results I suggest reporting only one digit for decimals in raw percentages.Table 3, probably even 1 digit for decimals for percentages is too much.

We appreciate the reviewer’s observation. Although in previous versions of our manuscript only one decimal was used for percentages, the authors decided to use a unified criteria in the representation of number through the manuscript, which for us gives continuity and coherence to our work’s presentation.

DiscussionThe first sentence states "in this longitudinal study". I am not sure this study design is really a longitudinal study, i.e. a cohort. Actually, the study seems to be analysed as repeated cross-sectional surveys or as several short follow up cohorts (one for each round). I do not think it is really important to give a name or classify the study design, but in the methods, you did not present it as a cohort, so you should not introduce this concept in the discussion.

We agree with the reviewer’s opinion on the classification of the study. Since it is a complex analysis, it can be difficult to classify it in the main and most common category designs. We have changed its denomination as a before-and-after study, which is the term also used in the abstract of the manuscript.

The following changes have been made to the manuscript.

Discussion (page 18):

“In this before-and-after study, we found that the pandemic reduced participation, but that this impact differed according to each woman’s history of participation. Although attendance was lower during the pandemic period, we found no significant differences in other performance indicators, such as the frequency of recall for additional tests after mammography, the percentage of false-positive results, or the cancer detection rate.”

Page 17, lines 19-25: I suggest also another possible explanation, not mutually exclusive to the one proposed by the authors. Among women not attending screening, there are probably many women who perform mammography more or less regularly in opportunistic screening with private providers. During the lockdown, all the providers experienced difficulties in organizing planned activity as mammography in asymptomatic women. Therefore, it is possible that women who usually perform mammography in private had difficulties in getting their mammogram timely and when received the invitation by the program decided to participate. This could be a specular effect of that observed in women regularly attending the screening program that this time decided not to attend, probably because the invitation was late and they seek for mammography elsewhere. The disruption of planned activities during the lockdown, in all providers, may explain both decrease in participation in regular attendees and the increase in never attenders. This is also consistent with a decrease in high socioeconomic status in participants post covid, in fact, these women are those who may have a higher propensity to seek a mammogram in the private sector.

We are very thankful for this very interesting addition to our discussion. It has been used to modify the discussion of our manuscript as addressed in the essential submissions section of this letter.

The authors correctly recognize the limit of assuming independence of the observation, but, if the id is woman-specific and is the same across the rounds, there is no need to make this assumption. I also agree that stratified analyses by type of round reduce the impact of this limit.

We thank the reviewer’s comments and have addressed this issue in the essential revisions section of this letter amongst similar comments in the methods section.

Page 19, line 1. "The same target population" this sentence may be confusing, the study does not have a closed cohort design, so each subsequent round targets slightly different populations.

We thank the reviewer for the observation and agree about the design of our study. Nevertheless, the target population is the same as in meaning that the criteria for the women who we consider part of this population does not change, although the women may enter and exit this target population as they stop meeting the inclusion criteria.

Reviewer #3 (Recommendations for the authors):I consider the work well done and interesting so for me is publishable with the following small revisions:1. The authors describe what invitation modes were used up to the pandemic (invitation letter with prefixed appointment) but it is not clear what recruitment strategies were used during the pandemic. Were there any changes in recruitment methods? It would be useful for the readers to get an idea.

We thank reviewer #3 for the recommendation and have already addressed the topic in the essential revisions section of this letter. Further information on the invitation process and changes has been provided and the manuscript has been modified accordingly.

2. Page 4, lines 25-32; wouldn't this whole part go into the results?

We thank the reviewer for the interesting opinion on the description of the population to be included in the Results section. Although this was a topic of discussion between the authors, we finally decided to include it in the “study population” subheading of the Methods section. The reason for this being that we wanted to keep the Results section as brief and concise as possible, with only the most relevant results of the study. We also felt that the population’s description could clarify the analyses and methods of the study.

3. One of the parameters used for the comparison is the recall rate. the authors limit their analysis the recall rate (number of women sent to the second level) and do not mention the compliance of women to in-depth diagnostics. Comparing the recall rates certainly allows to highlight a potential change in radiologists' behaviours, connected to all sanitization and social distancing measures that had to be put in place. It could have been interesting also to evaluate the presence (or not) of a change in women's behaviour and their willingness to compliance further assessments, despite the pandemic.

This is a very interesting point which we have addressed in the essential revisions section of this letter. We would like to thank the reviewer for the approach.

4. Among the analysed parameters, invitation coverage (the number of women invited out of the target population) was not included: it would have been interesting to know the trend of this parameter as an index of the resilience of the screening program.

All the women between 50 and 69 years of age living in the area of reference of our program, and who do not meet any exclusion criteria are invited to participate. Every year the dataset is actualized with the new women meeting inclusion criteria, whether because they turned 50 or because they moved to our area of reference. Although it is possible that some women are not correctly registered or that they live in our area but without it being their official residence, and we know that there are unregistered women living in our area, we do not have access to data on these women so it would not be possible for us to find the real denominator to see invitation coverage.

5. the authors made a pre and post covid-19 pandemic comparison. It is a very marginal issue but I would not talk about post covid since we are still inside the pandemic (although hopefully in the final stages); what about pre covid and covid?

We thank the reviewer for the suggestion as we had discussion within the study’s team on the nomenclature that best fitted our study. Since the greatest impact in preventive care was due to the interruption of the programs during the national lock-down between March and June 2020, “post-covid-19” actually means “post-lockdown”. It will be very hard to acknowledge the real impact of the whole COVID-19 since, as the reviewer commented, we are already living on it. Restrictions and incidence varied widely over the last two years so it would probably be unwise to assume that with such an early study, we can reflect on the pandemic’s global effect on breast cancer screening. Here, we meant only to study the effect that the cancelation and postponement of invitations and mammogram appointments could have affected the performance of our program. Despite this, we used the “pre/post- COVID-19” terminology to simplify the tables and text as it is a short and comprehensive nomenclature that briefly states the objective of the study.

References:

1. Amram O, Robison J, Amiri S, Pflugeisen B, Roll J, Monsivais P. Socioeconomic and Racial Inequities in Breast Cancer Screening during the COVID-19 Pandemic in Washington State. JAMA Netw Open. 2021;4(5):2019–22.

2. Miller MM, Meneveau MO, Rochman CM, Schroen AT, Lattimore CM, Gaspard PA, et al. Impact of the COVID-19 pandemic on breast cancer screening volumes and patient screening behaviors. Breast Cancer Res Treat [Internet]. 2021;189(1):237–46. Available from: https://doi.org/10.1007/s10549-021-06252-1

3. Mottram R, Knerr WL, Gallacher D, Fraser H, Khudairy LA-, Ayorinde A, et al. Factors associated with attendance at screening for breast cancer: a systematic review and meta- analysis. BMJ. 2021;11.

4. Posso M, Comas M, Román M, Domingo L, Louro J, González C, et al. Comorbidities and Mortality in Patients With COVID-19 Aged 60 Years and Older in a University Hospital in Spain. Arch Bronconeumol. 2020;56(11)(November):756–8.

5. Manzanares I, Sevilla Guerra S, Lombraña Mencía M, Acar-Denizli N, Miranda Salmerón J, Martinez Estalella G. Impact of the COVID-19 pandemic on stress, resilience and depression in health professionals: a cross-sectional study. Int Nurs Rev. 2021;(August 2020):1–10.

6. Haldane V, De Foo C, Abdalla SM, Jung AS, Tan M, Wu S, et al. Health systems resilience in managing the COVID-19 pandemic: lessons from 28 countries. Nat Med [Internet]. 2021;27(6):964–80. Available from: http://dx.doi.org/10.1038/s41591-021-01381-y

7. Rodriguez C, Plasencia A, Schroeder DG. Predictive factors of enrollment and adherence in a breast cancer screening program in Barcelona (Spain). Soc Sci Med. 1995 Apr 1;40(8):1155–60.

8. Song H, Bergman A, Chen AT, Ellis D, David G, Friedman AB, et al. Disruptions in preventive care: Mammograms during the COVID-19 pandemic. Health Serv Res. 2020;1–7.

9. Chen RC, Haynes K, Du S, Barron J, Katz AJ. Association of Cancer Screening Deficit in the United States with the COVID-19 Pandemic. JAMA Oncol. 2021;7(6):878–84.

10. Toyoda Y, Katanoda K, Ishii K, Yamamoto H, Tabuchi T. Negative impact of the COVID-19 state of emergency on breast cancer screening participation in Japan. Breast Cancer [Internet]. 2021;28(6):1340–5. Available from: https://doi.org/10.1007/s12282-021-01272-7

11. Giordano L, Castagno R, Giorgi D, Piccinelli C, Ventura L, Segnan N, et al. Breast cancer screening in Italy: Evaluating key performance indicators for time trends and activity volumes. Epidemiol Prev. 2015;39(3):30–9.

12. Castells X, Sala M, Ascunce N, Salas D, Zubizarreta R, Casamitjana M, et al. Descripción del cribado del cáncer en España: Proyecto DESCRIC. Informes de Evaluación de Tecnologías Sanitarias, AATRM núm. 2006/01. 2007. 1–327 p.

13. Posso M, Alcántara R, Vázquez I, Comerma L, Baré M, Louro J, et al. Mammographic features of benign breast lesions and risk of subsequent breast cancer in women attending breast cancer screening. Eur Radiol. 2022;32(1):621–9.

[Editors' note: further revisions were suggested prior to acceptance, as described below.]

The manuscript has been improved but there are some remaining issues that need to be addressed, as outlined below:1. The first paragraph of the Discussion should be modified to reflect the new results, as some of the changes in recall and false positives are now significant.

We thank the editors for bringing to our attention the incoherence of the first paragraph with our updated results. We proceeded to change the content of this paragraph in order to align it with our updated findings as seen below:

Discussion (page 14):

“In this before-and-after study, we found that the pandemic reduced participation, but also that this impact differed according to each woman’s history of participation. The frequency of recall for additional tests after mammography and the percentage of false-positive results were also significantly lower for incident screening, while for prevalent screening the reduction was only statistically significant for recall. Nevertheless, we found no significant differences in compliance with recall or cancer detection rate. “

2. The revision indicates that different invitation approaches were used for different types of participants (first, regular, non-participant) during the post-COVID era. Please discuss this in the Discussion, and how this may have impacted the results in Figure 1; it is likely some of the differences in participation may be due to these different invitation practices.

We would like to thank the editors for their interest on the changes in the invitation process. As explained in the previous response letter, the invitation process changed during the pandemic, depending on the previous participation history of each woman, in order to maximize the program’s efficiency. These changes may have affected participation regardless of the impact of the pandemic and may explain the differences seen between different screening groups. Nevertheless, the results are opposite to what we would expect from the changes in invitation. In contrast to the traditional pre-fixed appointment letter, regular participants and first-time invitees received a phone call to set an appointment for a mammogram. For previous non-participants, instead of a pre-fixed appointment, an open letter advising to call to make an appointment was sent. We presumed that the telephone call would engage more women but the groups with whom we used this strategy were those in which participation decreased more significantly. On the other hand, women without a mammography appointment of phone call seemed to participate more than before, despite still being the group with the smallest participation.

Despite these surprising results, we feel the implementation of phone calls as an invitation method actually increases participation, and we can only speculate that, using the previous invitation method, the participation decrease would have been even more aggravating. In fact, previous research in our screening program has shown that invitation through direct contact can increase participation, at least among lower socio-economic status. Telephone reminders have also proved to be an effective way to improve participation in different setting, although in most cases they follow a pre-set appointment.

With previous non-participants, the opposite seems to happen. Although previous evidence proves that open-letter invitations are lees effective, we noticed an increase in participation. We believe this increase may be related to other factors previously mentioned like increased health-consciousness and delays in private-care.

We have reflected these considerations in our discussion as follows:

Discussion (page 14):

“Since the invitation process was adapted to the pandemic, the changes in participation could be related to the different strategies used to invite previous participants and non-participants. Nevertheless, previous research in our program showed that participation increased with invitation through direct contact with women (24), which could be comparable to telephone calls. This effect seemed to be especially relevant in low socio-economic status areas, where there are more regular participants. Telephone reminders have also been proved to increase participation in different settings, although they usually follow a pre-set invitation date by postal mail (25). Considering this evidence, we would assume that, without the telephone call, the decrease in participation could have been greater among regular participants. For previous non-participants, it is unlikely that an open invitation letter instead of a pre-set appointment could increase participation. Research in cervical cancer screening has proved that pre-set appointments increase participation in comparison to open invitations (26), and therefore we believe that the increase in participation can be more feasibly explained by the previously discussed changes in private care and health-consciousness.”

3. The analysis of compliance with recall (Table 5 in the response to reviewers) would be of interest to some readers and should be included in the manuscript.

We thank the editors insight on the interest that the analysis of compliance with recall may have for readers and we have included it in the manuscript as requested. Some changes have been added to all section of the manuscript to reflect the addition of this outcome as follows:

Abstract (page 2):

“No significant differences were observed in compliance with recall (OR = 1.26, 95%CI 0.76-2.23), cancer detection rate (aOR=0.91 [95%CI=0.69-1.18]) or cancer stages.”

Methods (page 6):

“We used 5 main indicators of the program: participation, recall, false-positives, compliance with recall and detection rate. In addition, we also compared the following characteristics of the detected tumors, histology (invasive vs. in situ), tumor size, lymphatic invasion, the presence of metastases, and stage at diagnosis.”

Methods (page 6):

“Three other outcomes were analyzed using only the screening participants. The recall rate was estimated as the percentage of participants who were advised to undergo further assessment to rule out malignancy, whether non-invasive or invasive (ultrasound, tomosynthesis, contrast-enhanced mammography, biopsy, and/or others). False positives were estimated based on the percentage of women who underwent additional non-invasive or invasive assessments but who did not have a diagnosis of cancer after completion of additional examinations. The detection rate was the number of breast cancers detected at screening per 1000 participants. We calculated this rate, stratifying by type of breast cancer histology (i.e., the invasive or in situ cancer detection rate). Finally, compliance with recall was analyzed only among patients advised to undergo further assessment, the percentage of these patients who agreed to take additional tests in our facilities.”

Methods (page 7):

“For participation, we included the following variables in the model: type of screening round (prevalent vs. incident), age group, and socio-economic index. We found a strong interaction between COVID-19 and the type of screening round. Therefore, we created a new variable, which represented this interaction. Hence, the final models for participation differentiated four screening groups (prevalent-first-time invitee, prevalent-previous nonparticipant, incident-regular participant, and incident-irregular participant). We obtained crude results and adjusted by age and socioeconomic index. For compliance with recall we used logistic regression model to obtain crude Odds Ratios since we did not adjust for any variables due to the reduced sample.”

Results (page 10):

“Compliance with recall did not significantly change in the post-COVID-19 round (OR = 1.26, 95%CI 0.76-2.23) remaining stable with more than 97% of the women advised to undergo further assessment taking additional diagnostic tests in our facilities (Table 4).”

Discussion (page 14):

“Nevertheless, we found no significant differences in compliance with recall and cancer detection rate.”

Discussion (page 15):

“Despite the lower participation, the remaining performance indicators in our program did not seem to be negatively affected by the pandemic. Our results showed a statistically significant reduction in the recall rates of both prevalent and incident screening. These findings could be due to the increased workload caused by COVID-19 patients at our and many other hospitals, which strongly affected the radiology department in 2020 (30). We feared that repeat visits to the hospital might be perceived as increasing the risk of COVID-19 exposure, dissuading some women from undergoing any further testing after mammography. However, our data show that compliance with recall remained stable, indicating that women who chose to participate in the first place also engaged with further assessment when advised to do so.”

Additionally, table 4 has been changed to table 5 and a new table showing compliance with recall has been added to the manuscript as table 4.